# Med-Tuning: A New Parameter-Efficient Tuning Framework for Medical Volumetric Segmentation

**Jiachen Shen**[*1]                                                        M202110559@XS.USTB.EDU.CN
**Wenxuan Wang**[*1]                                                       S20200579@XS.USTB.EDU.CN
[1] *School of Automation and Electrical Engineering, University of Science and Technology Beijing*
**Chen Chen**[2]                                                          CHEN.CHEN@CRCV.UCF.EDU
[2] *Center for Research in Computer Vision, University of Central Florida*
**Jianbo Jiao**[3]                                                        J.JIAO@BHAM.AC.UK
[3] *School of Computer Science, University of Birmingham*
**Jing Liu**[4]                                                           JLIU@NLPR.IA.AC.CN
[4] *Institute of Automation, Chinese Academy of Sciences*
**Yan Zhang**[1]
**Shanshan Song**[1]
**Jiangyun Li**[†1]                                                       LEEJY@USTB.EDU.CN

**Editors:** Accepted for publication at MIDL 2024

## Abstract

The "pre-training then fine-tuning (FT)" paradigm is widely adopted to boost the model performance of deep learning-based methods for medical volumetric segmentation. However, conventional full FT incurs high computational and memory costs. Thus, it is of increasing importance to fine-tune pre-trained models for medical volumetric segmentation tasks in a both effective and parameter-efficient manner. In this paper, we introduce a new framework named Med-Tuning to realize parameter-efficient tuning (PET) for medical volumetric segmentation task and an efficient plug-and-play module named Med-Adapter for task-specific feature extraction. With a small number of tuned parameters, our framework enhances the 2D baselines's precision on segmentation tasks, which are pre-trained on natural images. Extensive experiments on three benchmark datasets (CT and MRI modalities) show that our method achieves better results than previous PET methods on volumetric segmentation tasks. Compared to full FT, Med-Tuning reduces the fine-tuned model parameters by up to 4×, with even better segmentation performance. Our project webpage is at https://rubics-xuan.github.io/Med-Tuning/.

**Keywords:** Parameter-Efficient Tuning, Medical Volumetric Segmentation, Transformer.

## 1. Introduction

Medical volumetric segmentation (MVS) task is to identify tumors and organ sub-regions in biomedical images, aiding accurate clinical diagnoses and treatment planning. It is crucial in medical research, due to the widespread use of 3D imaging like computed tomography (CT) and magnetic resonance imaging (MRI). In the last decades, a large number of deep neural network architectures have been proposed, including convolutional neural networks (CNNs) (e.g., (Milletari et al., 2016; Çiçek et al., 2016; Isensee et al., 2021)) and Transformer-based networks (e.g., (Cao et al., 2022; Hatamizadeh et al., 2022b,a;

---

[*] Contributed equally
[†] Corresponding author

Zhou et al., 2023; Peiris et al., 2022)). Recently, the "pre-training then fine-tuning" paradigm (Yosinski et al., 2014) has gained much popularity to enhance model performance in downstream tasks. As in (Cao et al., 2022), the conventional full fine-tuning scheme updates all parameters of the pre-trained models. Yet, as models continuously improve in performance, particularly Transformer-based ones like (Cao et al., 2022; Hatamizadeh et al., 2022b,a), their tuned parameter count escalates significantly. Thus, full fine-tuning involves a lot of tuned parameters and entails great training costs. To reduce tuned parameters, head-tuning (Head) was proposed (He et al., 2022), focusing solely on optimizing the task-specific decoder, albeit resulting in decreased model performance. Meanwhile, recent studies (Jia et al., 2022; Chen et al., 2022; Pan et al., 2022; Sung et al., 2022; Yu et al., 2022; Zhang et al., 2023; Xu et al., 2023; Wu et al., 2023; Fischer et al., 2024) focus on parameter-efficient tuning (PET) to balance model performance and tuned parameters.

In this paper, we aim to investigate how to adapt strong visual foundation models pre-trained on natural images to MVS tasks via PET. We initiate our analysis with some examples of widely available models that use image-level pre-training (e.g., classification task (Deng et al., 2009), CLIP (Radford et al., 2021), MOCO v3 (Chen et al., 2021)) in natural image domain. Figure 1 presents the two-fold gaps between upstream pre-training and downstream fine-tuning:

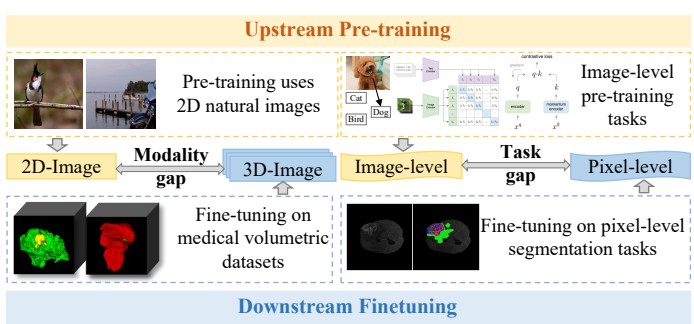

Figure 1: The two-fold gaps between upstream pre-training and our downstream fine-tuning.

**(1) Domain gap** between natural images and medical volumes; **(2) Task gap** between image-level pre-training and pixel-level segmentation. To narrow these gaps, we propose **Med-Tuning**, a new PET framework for MVS, and **Med-Adapter**, an efficient plug-and-play module for task-specific feature extractions. Med-Tuning processes 3D volumes through a frozen pre-trained Transformer model with inserted Med-Adapters. Med-Adapter greatly narrow both gaps by capturing spatial multi-scale features and volumetric correlations between slices with few additional parameters. Our main contributions are summarized as follows:

- We propose a new PET framework **Med-Tuning**, which greatly boosts the performance of the pre-trained models on MVS task and reduces training costs.
- We propose a plug-and-play module **Med-Adapter**, to consider both spatial relationship modeling (coarse/fine-grained) and volumetric correlations between slices.
- Extensive experiments on three benchmark datasets with both CT and MRI modalities convince the effectiveness of **Med-Tuning** over full fine-tuning and other PET methods.
- **Med-Tuning** adapts well to the rapidly evolving Transformer-based visual foundation models (i.e., SAM), showcasing strong generalization and flexibility.

## 2. Related Work

### 2.1. Medical Volumetric Segmentation

Achieving promising performance on MVS requires the incorporation of both spatial multi-scale representations and volumetric correlations, as demonstrated by prior research (Hatamizadeh

et al., 2022a). Several U-Net inspired CNN-based models (Ronneberger et al., 2015; Çiçek et al., 2016; Zhou et al., 2018; Isensee et al., 2021) concatenate multi-scale features from the encoder and up-sampled features, complementing the loss of spatial information caused by down-samplings. Cao et al. (Cao et al., 2022) use skip connections to effectively fuse low-level and high-level features in Transformers. Besides, various MVS methods capture volumetric correlations by 3D convolutions (Çiçek et al., 2016; Milletari et al., 2016; Isensee et al., 2021) or self-attention mechanism among 3D volumes (Wang et al., 2021).

### 2.2. Visual Parameter-Efficient Tuning

Recently, novel vision PET methods have emerged to balance accuracy and tuned parameter efficiency during fine-tuning, which can be categorized into three groups: **(1)** Prompt-based methods (Zhang et al., 2023; Fischer et al., 2024). For instance, VPT (Jia et al., 2022) adds learnable prompt tokens to patch embeddings for downstream visual tasks. Pro-tuning (Nie et al., 2023) inserts multiple stage-wise prompt blocks into different stages of the backbone. **(2)** Adapter-based methods (Houlsby et al., 2019; Chen et al., 2022; Yang et al., 2023; Wu et al., 2023). Adapter is a lightweight module inserted between the feed-forward layer and layer normalization in Transformer, which are tuned during fine-tuning while other layers stay frozen. ST-Adapter (Pan et al., 2022) introduces 3D depth-wise convolution (DWConv) (Ye et al., 2019) in Adapter modules to capture spatial-temporal features. **(3)** Other PET techniques. SAN (Xu et al., 2023) is a small and separate network that is trained via shortcut connections from backbone to reduce memory cost during fine-tuning. Recent studies (Wu et al., 2023; Chai et al., 2023) mainly focus on exploring the potential of the Segment Anything Model (SAM) for medical image analysis.

### 2.3. Utilization of Fourier Transform in Computer Vision

Image analysis in Fourier domain is extensively used in various vision tasks (Ding et al., 2017; Lee et al., 2018; Li et al., 2020; Chi et al., 2020; Yang and Soatto, 2020; Rao et al., 2021). Fast Fourier Transform (FFT) and Inverse Fast Fourier Transform (IFFT) leverage frequency information for global connectivity through parameter-free domain mapping on original images, resulting in an intrinsic global vision characteristic. According to the conclusion of (Oppenheim et al., 1979; Liu et al., 2023a), the phase-only image or feature retains many of the semantics features of the original image.

## 3. Methodology

### 3.1. Preliminaries

**Vanilla Adapter.** Given an input feature $X \in \mathbb{R}^{N \times d}$, the vanilla Adapter can be represented as Equation (1) (Houlsby et al., 2019) , where $W_{down}$ and $W_{up}$ indicate the down-projection and up-projection layer, $\sigma(\cdot)$ is an activation function, $+$ is a skip-connection.

$$\text{Adapter}(X) = X + \sigma(XW_{down})W_{up}, \tag{1}$$

**Discrete Fourier Transform.** Discrete Fourier Transform (DFT) serves as classical technique for computer vision applications (Rao et al., 2021). Given a 3D input (volumetric data or feature) $x[D, H, W]$, DFT is defined as:

$$X = \mathcal{F}(x) = \sum_{w=0}^{W-1} \sum_{h=0}^{H-1} \sum_{d=0}^{D-1} x(d, h, w) e^{-j2\pi\left(\frac{xd}{D} + \frac{yh}{H} + \frac{zw}{W}\right)} = \mathcal{R} + \mathcal{I}j, \tag{2}$$

where $X$ is a complex matrix, $\mathcal{R}$ and $\mathcal{I}$ denote its real and imaginary part. In implementation, we use the accelerated versions of DFT and Inverse DFT (i.e., FFT and IFFT).

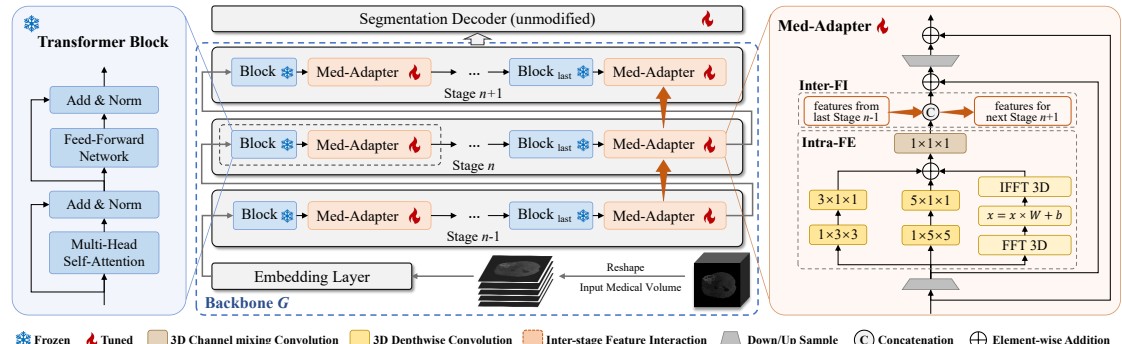

Figure 2: The overall architecture of Med-Tuning. Med-Tuning consists of a 2D Transformer baseline with proposed Med-Adapters inserted at each encoder stage. Only Med-Adapters and decoder are tuned while all the other layers stay frozen.

## 3.2. Med-Tuning: Parameter-Efficient Tuning for MVS

The overall architecture of our framework is depicted in Figure 2. Med-Tuning consists of a 2D Transformer backbone $G$ pre-trained on natural images, a segmentation decoder and inserted Med-Adapter. Given a batch of medical volumes input $I[B, C, D, H, W]$, $(B, C, D, H, W$ is the number of batch size, channel, slice, height, width), we initially reshape them to $I[BD, C, H, W]$ before embedding layer. According to two main considerations in the following, we decide to exclusively introduce Med-Adapter into encoder without modifying decoder, enabling high decoder scalability to meet different requirements. First, encoder plays a pivotal role in baseline. Inadequate feature extraction will hinder performance even with the same robust decoder, as evidenced by the decline in results for Head compared to Full, detailed in Table 1, Table 2 and Table 3. Besides, not all segmentation decoders come with pre-trained weights, necessitating the fine-tuning of the entire decoder. Secondly, sole insertion in encoder part improves the flexibility of the whole framework. Our inserting strategy broadens the adaptability of Med-Tuning on visual foundation models while reducing tuned parameters.

## 3.3. Med-Adapter: Adapter for MVS

We propose a task-oriented and simple yet effective module for medical volumetric data, namely **Med-Adapter**. The motivation of Med-Adapter is to empower a 2D Transformer model pre-trained on natural images to gain the capability of volumetric feature modeling in a parameter-efficient manner. Here we consider three criteria when designing Med-Adapter: (1) *MVS oriented*: It's necessary to narrow the mentioned gaps in Figure 1. (2) *Light-weight*: Structure with a low amount of parameters is crucial. (3) *Plug-and-play*: An easy-to-implement module is friendly to practical deployment. While retaining the bottleneck structure of the vanilla Adapter (Equation (1)), we introduce a few tailored designs into Med-Adapter based on above criteria, shown in Figure 2 (right).

### 3.3.1. Intra-stage Feature Enhancement (Intra-FE).

We introduce multiple branches tailored to capture fine-grained feature representations, coarse-grained global semantics and volumetric correlations among slices, which are vital for realizing accurate medical volumetric segmentation task.

**Multi-scale Local Branch (MLB)** We employ parallel 3D convolutions with kernels of sizes $k = 3$ and $k = 5$ to extract spatial multi-scale features and 3D volumetric correlations between slices. The conventional 3D convolution operations are replaced with 3D DWConv in a parameter-efficient manner. Moreover, we use two cascaded $1 \times k \times k$ and $k \times 1 \times 1$ convolutions to replace the $k \times k \times k$ convolution to pursue an extremely lightweight structure.

**FFT Global Branch (FGB)** To achieve coarse-grained global semantic extraction in a parameter-efficient way, we substitute traditional large convolutional kernels and attention mechanisms, known for their memory and computation demands, with 3D FFT, IFFT and learnable complex matrices. These filter-like complex matrices are designed to model frequency features that contain global semantics in the whole frequency domain. Compared to vanilla 3D self-attention operation with $\mathcal{O}(n^3)$ complexity ($n$ is the number of tokens), our FGB is a lightweight module. In detail, the computational complexity of FGB is $\mathcal{O}(nlog(n))$, where FFT and IFFT are both with $\mathcal{O}(nlog(n))$ complexity while that of Hadamard product or matrix addition is $\mathcal{O}(n)$. Then we merge the aforementioned branches by efficient $1 \times 1 \times 1$ convolution. Therefore, given the intermediate embedded feature representations $X \in [BD, C', HW]$, the Intra-FE module can be expressed as Equation (3)-(5). In this way, the Intra-FE module is theoretically capable of modeling volumetric correlations among slices and incorporating abundant spatial multi-scale features for the downstream dense prediction task (i.e., MVS).

$$\text{Intra-FE} = M = Conv_{1 \times 1 \times 1}(\text{MLB} + \text{FGB}), \tag{3}$$

$$\text{MLB} = DWConv_3(X') + DWConv_5(X'), \ \text{FGB} = \mathcal{F}^{-1}(W_F \odot \mathcal{F}(X') + b_F) \tag{4}$$

$$X' = \text{R}_{volume}(\sigma(XW_{down})), \ X' \in [B, C', D, H, W], \ C' = C/\alpha \tag{5}$$

where $M$ is the intra-stage enhanced feature representations. $W_F$ and $b_F$ are the learnable complex matrices, $\mathcal{F}$ is 3D FFT in Equation (2) and $\mathcal{F}^{-1}$ is 3D IFFT. $\odot$ is the hadamard product, while $\text{R}_{volume}$ is a reshape operation to obtain cube-shape feature representations.

### 3.3.2. Inter-stage Feature Interaction (Inter-FI).

To fully exploit the representations collected by Inter-FI modules in Med-Adapter at each stage, we further consider the feature interaction between different stages. The intra-stage enhanced feature representations $M$ in this stage will be fused with $M_{LastStage}$ (i.e., the output of the Intra-FE module from the previous stage). Note that Inter-FI is only introduced at specific Med-Adapters at the end of each stage. Thus, Inter-FI is expressed as Equation (6). In this way, feature representations extracted by Intra-FE modules of Med-Adapters in shallow layers are gradually fed to adjacent higher layers, realizing inter-stage feature interaction by explicit enhancement and thus boosting model performance.

$$\text{Inter-FI} = Cat(\mathcal{A}(H, M_{LastStage})), \ M = \begin{cases} \text{Inter-FI}, & \text{if } Flag_{last} \\ M, & \text{if not } Flag_{last} \end{cases} \tag{6}$$

where $\mathcal{A}$ denotes convolution operations to align $M$ and $M_{LastStage}$ in terms of spatial resolution and channel dimension, $Cat$ refers the concatenation. $Flag_{last}$ is a bool parameter and $Flag_{last} = True$ when the current Med-Adapter is the inserted last one at this stage.

In summary, our Med-Adapter can be formulated as Equation (7). $\text{R}_{flatten}$ denotes a symmetric operation that reshapes the feature back to the original shape of $X$.

$$\text{Med-Adapter}(X) = X + \text{R}_{flatten}(M + \text{R}_{volume}(\sigma(XW_{down}))W_{up}. \tag{7}$$

## 4. Experiments and Results

### 4.1. Experimental Setup

**Datasets and Evaluation Metrics.** Our proposed Med-Tuning is evaluated on three benchmark datasets: **(1)** Kidney Tumor Segmentation 2019 (Heller et al., 2019)(KiTS 2019), **(2)** Brain Tumor Segmentation 2019 (BraTS 2019) (Menze et al., 2014; Bakas et al., 2017, 2018), **(3)** Brain Tumor Segmentation 2020 (BraTS 2020) (Menze et al., 2014; Bakas et al., 2017, 2018), detailed in Appendix A.1. KiTS 2019 dataset comprises multi-phase 3D CTs depicting the kidneys and tumors. The ground truth contains 3 classes: background (label 0), kidney (label 1), and kidney tumor (label 2). The segmentation accuracy of KiTS 2019 is measured by kidney dice (label 1 and 2) and tumor dice (label 2), composite dice (the average of kidney and tumor dice). BraTS 2019 and BraTS 2020 datasets consist of 3D brain MRI scans with four modalities. The ground truth contains 4 classes: background (label 0), necrotic and non-enhancing tumor (label 1), peritumoral edema (label 2), and GD-enhancing tumor (label 4). The segmentation accuracy is measured by Dice score and the Hausdorff distance (95%) metrics for enhancing tumor region (ET, label 4), regions of tumor core (TC, labels 1 and 4), and whole tumor region (WT, labels 1,2 and 4).

**Implementation Details.** Experiments utilizing PyTorch (Paszke et al., 2019) for implementation are conducted on NVIDIA GeForce RTX 3090 GPUs. The pre-trained vanilla ViT (Dosovitskiy et al., 2021) with UPerNet (Xiao et al., 2018) decoder (ViT+UPerNet) and Swin-UNet (Cao et al., 2022) based on pre-trained Swin Transformer(tiny) are our chosen baselines. More implementation details are available in Appendix A.2.

### 4.2. Results and Analysis

We conduct experiments on three benchmark validation sets and compare our method with scratch (i.e., training with random initialization, without pre-training), full fine-tuning, head tuning and previous state-of-the-art PET approaches (i.e., VPT (Jia et al., 2022), Adapter (Houlsby et al., 2019), AdaptFormer (Chen et al., 2022), Pro-tuning (Nie et al., 2023), ST-Adapter (Pan et al., 2022)). Qualitative results are shown in Appendix B.

**KiTS 2019.** The performance comparisons with ViT+UPerNet and Swin-UNet as baseline are shown in Table 1. Our proposed method boosts the performance of full fine-tuning considerably and achieves much higher Dice scores than previous PET methods, with much fewer tuned model parameters. In comparison with recently proposed PET methods (e.g., VPT, Pro-tuning and ST-Adapter), our Med-Tuning achieves better performance-efficiency trade-off on two baselines. Specifically, Med-Tuning improves model performance by a large margin (i.e., ↑ **4.20%** Kidney Dice, ↑ **17.13%** Tumor Dice, ↑ **10.67%** Composite Dice on ViT+UPerNet and ↑ **1.01%** Kidney Dice, ↑ **8.02%** Tumor Dice, ↑ **4.52%** Composite Dice on Swin-UNet) with only **17.70%** and **27.58%** of tuned parameters respectively in comparison with full fine-tuning.

**BraTS 2019.** Performance comparisons on BraTS 2019 on two baselines are shown in Table 2 (left) and Table 3 (left). Results show that our method attains the best trade-off between performance and efficiency, achieving comparable or even better results than previous methods. Compared to full fine-tuning, Med-Tuning achieves maximum improvements of **4.23%** (ViT+UPerNet) and **1.28%** (Swin-UNet) in Dice scores. Our Med-Tuning also achieves high parameter efficiency, tuning only **17.70%** parameters of ViT and **27.58%** of Swin-UNet, with inserted parameters being only **2.82%** of ViT and **2.72%** of Swin-UNet.

Table 1: Performance comparison on KiTS 2019 with Swin-UNet and ViT+UPerNet. Blue and Green text denote the percentage of tuned parameters and the performance improvement compared to full fine-tuning (with grey background).

| KiTS 2019 | ViT+UPerNet | | | | | Swin-UNet | | | | |
| --- | --- | --- | --- | --- | --- | --- | --- | --- | --- | --- |
| | Tuned Params(M) | Inserted Params(M) | Dice (%) ↑ | | | Tuned Params(M) | Inserted Params(M) | Dice (%) ↑ | | |
| | | | Kidney | Tumor | Composite | | | Kidney | Tumor | Composite |
| Scratch | 100.849 | - | 88.01 | 46.53 | 67.27 | 27.154 | - | 94.33 | 61.10 | 77.71 |
| Full | 100.849 | - | 87.32 | 47.34 | 67.33 | 27.154 | - | 94.68 | 62.13 | 78.40 |
| Head | 15.007 | - | 87.35 | 42.85 | 65.10 | 6.752 | - | 91.95 | 53.93 | 72.94 |
| VPT-Shallow | 15.015 | 0.008 | 86.91 | 41.67 | 64.29 | 6.753 | 0.001 | 91.72 | 54.86 | 73.29 |
| VPT-Deep | 15.100 | 0.092 | 88.01 | 46.45 | 67.23 | 6.780 | 0.029 | 91.53 | 53.41 | 72.47 |
| Adapter | 18.567 | 3.560 | 89.75 | 49.03 | 69.39 | 7.541 | 0.790 | 93.02 | 57.15 | 75.08 |
| AdaptFormer | 16.197 | 1.190 | 87.62 | 44.46 | 66.04 | 7.124 | 0.372 | 93.74 | 59.79 | 76.77 |
| Pro-tuning | 19.812 | 4.805 | 89.44 | 48.32 | 68.88 | 8.359 | 1.607 | 90.34 | 51.19 | 70.77 |
| ST-Adapter | 22.118 | 7.110 | 90.33 | 61.29 | 75.81 | 8.328 | 1.577 | 92.97 | 57.33 | 75.15 |
| **Ours** | 17.853 | 2.846 | **91.52** | **64.47** | **78.00** | 7.489 | 0.738 | **95.69** | **70.14** | **82.92** |
| | 17.70% | 2.82% | (+4.20) | (+17.13) | (+10.67) | 27.58% | 2.72% | (+1.01) | (+8.01) | (+4.52) |

Table 2: Performance comparison on BraTS 2019 and BraTS 2020 with ViT+UPerNet.

| ViT+UPerNet | Tuned Params (M) | Inserted Params (M) | BraTS 2019 | | | | | | BraTS 2020 | | | | | |
| --- | --- | --- | --- | --- | --- | --- | --- | --- | --- | --- | --- | --- | --- | --- |
| | | | Dice (%) ↑ | | | Hausdorff (mm) ↓ | | | Dice (%) ↑ | | | Hausdorff (mm) ↓ | | |
| | | | ET | WT | TC | ET | WT | TC | ET | WT | TC | ET | WT | TC |
| Scratch | 100.849 | - | 64.96 | 83.03 | 71.34 | 7.64 | 10.60 | 10.94 | 65.80 | 83.72 | 72.01 | 32.48 | 10.06 | 21.47 |
| Full | 100.849 | - | 68.49 | 85.56 | 75.12 | 6.67 | 7.88 | 10.53 | 69.12 | 85.90 | 75.29 | 34.43 | 7.32 | 17.09 |
| Head | 15.007 | - | 65.71 | 84.19 | 74.77 | 6.13 | 7.51 | 7.86 | 66.03 | 84.50 | 74.47 | 37.81 | 7.47 | 14.15 |
| VPT-Shallow | 15.015 | 0.008 | 66.02 | 84.72 | 75.84 | 6.11 | 7.51 | 8.47 | 66.52 | 84.82 | 75.46 | 37.77 | 7.47 | 13.53 |
| VPT-Deep | 15.100 | 0.092 | 67.01 | 85.14 | 76.80 | 6.06 | 7.72 | 7.65 | 67.69 | 85.28 | 76.59 | 31.77 | 7.74 | **10.62** |
| Adapter | 18.567 | 3.560 | 68.30 | 85.37 | 77.05 | **5.50** | 7.64 | 7.99 | 68.58 | 85.77 | 77.00 | 32.63 | 8.17 | 16.18 |
| AdaptFormer | 16.197 | 1.190 | 65.88 | 84.34 | 74.77 | 6.65 | 8.20 | 8.43 | 65.52 | 84.14 | 74.28 | 41.03 | 8.39 | 14.78 |
| Pro-tuning | 19.812 | 4.805 | 67.18 | 85.32 | 76.51 | 5.81 | 7.07 | 7.56 | 67.28 | 85.57 | 76.58 | 40.43 | 7.00 | 12.87 |
| ST-Adapter | 22.118 | 7.110 | 69.18 | 86.27 | 79.18 | 6.08 | 6.94 | **6.78** | 68.60 | 86.55 | **79.52** | 34.06 | 6.79 | 12.77 |
| **Ours** | 17.853 | 2.846 | **70.53** | **86.58** | **79.35** | 5.86 | **6.22** | 6.95 | **70.69** | **86.69** | 79.36 | **28.64** | **6.20** | 15.05 |
| | 17.70% | 2.82% | (+2.04) | (+1.02) | (+4.23) | (-0.81) | (-1.66) | (-3.58) | (+1.57) | (+0.79) | (+4.07) | (-5.79) | (-1.12) | (-2.04) |

**BraTS 2020.** Performance comparisons on BraTS 2020 are shown in Table 2 (right) and Table 3 (right). Compared to full fine-tuning, Med-Tuning achieves maximum improvements of **4.07%** (ViT+UPerNet) and **1.64%** (Swin-UNet) in Dice scores with very few tuned parameters, surpassing most of PET methods. Compared to ST-Adapter, our tuned parameters are fewer yet yield a more substantial overall performance improvement. Moreover, Med-Tuning took about 1.34 (ViT+UPerNet) and 1.68 (Swin-UNet) hours for fine-tuning, 0.76 (ViT+UPerNet) and 0.46 (Swin-UNet) minutes per sample for inference.

Table 3: Performance comparison on BraTS 2019 and BraTS 2020 with Swin-UNet.

| Swin-UNet | Tuned Params (M) | Inserted Params (M) | BraTS 2019 | | | | | | BraTS 2020 | | | | | |
| --- | --- | --- | --- | --- | --- | --- | --- | --- | --- | --- | --- | --- | --- | --- |
| | | | Dice (%) ↑ | | | Hausdorff (mm) ↓ | | | Dice (%) ↑ | | | Hausdorff (mm) ↓ | | |
| | | | ET | WT | TC | ET | WT | TC | ET | WT | TC | ET | WT | TC |
| Scratch | 27.154 | - | 78.38 | 88.59 | 76.46 | 6.06 | 10.65 | 9.18 | 78.72 | 89.12 | 77.07 | 7.62 | 6.98 | 19.08 |
| Full | 27.154 | - | 78.26 | 89.56 | 79.16 | 4.33 | 6.15 | 6.70 | 79.09 | 89.87 | 79.15 | 9.67 | 6.03 | 15.31 |
| Head | 6.752 | - | 78.07 | 88.68 | 77.26 | 5.02 | 6.70 | 7.09 | 78.77 | 88.66 | 76.90 | **4.89** | 8.49 | 16.06 |
| VPT-Shallow | 6.753 | 0.001 | 77.16 | 88.30 | 76.77 | 5.42 | 6.15 | 7.35 | 77.43 | 88.23 | 76.13 | 7.53 | 6.07 | 16.07 |
| VPT-Deep | 6.780 | 0.029 | 77.02 | 88.65 | 76.91 | 5.30 | 7.09 | 7.94 | 78.63 | 88.80 | 77.17 | 8.27 | 6.23 | 13.25 |
| Adapter | 7.541 | 0.790 | 77.98 | 89.22 | 78.02 | 5.30 | 6.62 | 8.49 | 78.51 | 89.16 | 77.71 | 7.05 | 6.25 | 19.09 |
| AdaptFormer | 7.124 | 0.372 | 77.69 | 88.61 | 76.83 | 4.91 | 6.29 | 7.89 | 78.22 | 88.92 | 76.40 | 10.35 | 6.48 | 16.90 |
| Pro-tuning | 8.359 | 1.607 | **78.58** | 89.33 | 78.79 | 5.27 | 6.41 | 8.24 | 78.77 | 89.46 | 78.20 | 7.31 | 6.50 | **10.54** |
| ST-Adapter | 8.328 | 1.577 | 78.40 | 89.54 | 77.44 | 4.75 | 6.01 | 7.41 | 78.96 | 89.54 | 77.85 | 7.67 | **5.48** | 15.53 |
| **Ours** | 7.489 | 0.738 | 78.51 | **89.68** | **80.44** | **4.00** | **5.52** | **5.76** | **79.25** | **90.06** | **80.79** | 12.40 | 4.41 | 11.59 |
| | 27.58% | 2.72% | (+0.25) | (+0.12) | (+1.28) | (-0.33) | (-0.63) | (-0.94) | (+0.16) | (+0.19) | (+1.64) | (+2.73) | (-1.62) | (-3.72) |

### 4.3. Ablation Studies

Extensive ablation experiments are conducted based on five-fold cross-validation. For more ablation experiments please refer to Appendix C.

**Inserted Position of Med-Adapter.** We conduct experiments on BraTS 2019 training set to assess the segmentation performance by inserting Med-Adapter at various stages of Swin-UNet encoder. Given that Swin-UNet encoder

Table 4: Ablation study on the position of inserted Med-Adapter.

| Encoder | | | | Dice (%) ↑ | | | HF (mm) ↓ | | |
|---|---|---|---|---|---|---|---|---|---|
| $n=0$ | $n=1$ | $n=2$ | $n=3$ | ET | WT | TC | ET | WT | TC |
| - | - | - | - | 78.07 | 88.68 | 77.26 | 5.02 | 6.70 | 7.10 |
| ✓ | - | - | - | - | - | - | - | - | - |
| ✓ | ✓ | - | - | 74.83 | 87.09 | 72.94 | 7.26 | 13.12 | 10.17 |
| ✓ | ✓ | ✓ | - | 75.60 | 86.79 | 73.41 | 8.44 | 12.32 | 11.24 |
| ✓ | ✓ | ✓ | ✓ | **78.51** | **89.68** | **80.44** | **4.00** | **5.52** | **5.76** |

has four continuous stages ($n = 0, 1, 2, 3$). According to Table 4, Inserting Med-Adapter in the initial stages resulted in degraded performance, with none surpassing our best default setting (gray background). This may be attributed to the greater contribution of features learned in later encoder stages when transferring pre-trained weights to the MVS task.

**Generalization Capability on Other Pre-trained Weights.** To explore the potential of our Med-Tuning, we investigate the effect of diverse encoder pre-trained weights (e.g., multi-modal based (CLIP (Radford et al., 2021)), self-supervised based (MAE (He et al., 2022), MoCo v3 (Chen et al., 2021)) and SAM (Kirillov et al., 2023)) on BraTS 2019 training set with ViT-B/16. As presented in Table 5, given different pre-trained weights, our easy-to-integrate framework boosts the performance consistently with much fewer tuned parameters, suggesting the effectiveness and robustness of our Med-Tuning framework.

**Generalization Capability on 3D Baseline and Medical Pre-trained Weight.** To demonstrate the generalization capability of our approach, we select Swin UNETR (Tang et al., 2022) pre-trained on medical datasets as a supplementary 3D baseline and experiment on part of the Medical Segmentation Decathlon (MSD) (Antonelli et al., 2022) dataset. For implementation details please refer to Appendix A.3. Experimental results in Table 6 show that our method still outperforms full fine-tuning in Memory, Time and Dice score.

Table 5: Ablations on other pre-trained weights.

| Pre-trained Weights | Method | Dice (%) ↑ | | |
|---|---|---|---|---|
| | | ET | WT | TC |
| CLIP | Full | 64.58 | 84.69 | 73.31 |
| | **Ours** | **68.05** | **86.29** | **77.34** |
| MAE | Full | 64.86 | 84.71 | 73.95 |
| | **Ours** | **66.32** | **85.50** | **78.05** |
| MoCo v3 | Full | 65.06 | 84.30 | 73.51 |
| | **Ours** | **67.09** | **85.45** | **77.41** |
| SAM | Full | 65.89 | 85.32 | 74.05 |
| | **Ours** | **67.64** | **86.10** | **78.33** |

Table 6: Ablations on MSD dataset with pre-trained Swin UNETR.

| Organ | Method | Memory(GB)↓ | Time(h)↓ | Dice_AVG(%)↑ |
|---|---|---|---|---|
| Task02 Heart (MRI) | Scratch | 19.73 | 1.05 | 91.95 |
| | Full | 19.73 | 1.06 | 93.73 |
| | **Ours** | **13.44** | **0.86** | **95.84** |
| Task06 Lung (CT) | Scratch | 23.51 | 8.39 | 65.82 |
| | Full | 23.51 | 8.39 | 67.69 |
| | **Ours** | **20.30** | **8.03** | **78.09** |
| Task09 Spleen (CT) | Scratch | 20.32 | 3.21 | 95.76 |
| | Full | 20.32 | 3.21 | 96.52 |
| | **Ours** | **19.71** | **2.22** | **97.06** |

## 5. Conclusion

In this work, we present a new PET framework named Med-Tuning with strong generalization capabilities for the practical application of MVS. Taking advantage of both spatial relationship modeling (coarse/fine-grained) and volumetric correlations, our framework achieves better volumetric segmentation accuracy on 2D baselines pre-trained on relatively easily acquired natural images. To some extent, Med-Tuning could consistently and sustainably boost the segmentation performance of pre-trained models on MVS tasks, keeping pace with the rapid development of foundation models in computer vision field.

## Acknowledgments

Jianbo Jiao is supported by the Royal Society grants IES\R3\223050 and SIF\R1\231009.

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

## Appendix A. Implementation Details.

### A.1. Details about Benchmark Datasets

Details about BraTS 2019, BraTS 2020 and KiTS 2019 datasets are shown in Table 7.

### A.2. Details about Benchmark Validation Experiments.

We employ pre-trained weights from two exemplary Transformer-based backbones, Swin Transformer tiny (Liu et al., 2021) pre-trained on ImageNet-1k and Vision Transformer base version (ViT-B/16) (Dosovitskiy et al., 2021) pre-trained on ImageNet-21k (Deng et al.,

Table 7: Details about BraTS 2019, BraTS 2020 and KiTS 2019 datasets.

| Dataset | Modality | Number of Training Cases | Number of Test Cases | Spatial Resolution |
|---|---|---|---|---|
| KiTS 2019 | CT | 210 | 90 | $512 \times 512$ |
| BraTS 2019 | MRI | 335 | 125 | $240 \times 240 \times 155$ |
| BraTS 2020 | MRI | 369 | 125 | $240 \times 240 \times 155$ |

2009). Swin-UNet (Cao et al., 2022) and ViT (Dosovitskiy et al., 2021) with UPerNet (Xiao et al., 2018) decoder (ViT+UPerNet) are chosen as two robust baselines to ensure equitable comparison. As shown in Table 8, the specific implementation details on BraTS 2019, BraTS 2020, and KiTS 2019 datasets for two baselines are comprehensively illustrated. On all three benchmark datasets, models are fine-tuned with a batch size of 16 and the Adam optimizer.

During training, the following data augmentation techniques are applied to BraTS 2019 and BraTS 2020 datasets: (1) random cropping from $240 \times 240 \times 155$ to $128 \times 128 \times 128$ voxels; (2) random mirror flipping across the axial, coronal and sagittal planes by a probability of 0.5; (3) random intensity shift between [-0.1, 0.1] and scale between [0.9, 1.1]. $L2$ Norm is also applied for regularization with a weight decay rate of $10^{-5}$. As for the KiTS 2019 dataset, the employed data augmentations follow as the prior work (Isensee et al., 2021).

Table 8: Implementation details on BraTS 2019, BraTS 2020 and KiTS 2019 datasets for two baselines (i.e., Swin-UNet, ViT+UPerNet).

| Dataset | Baseline | Backbone | Pre-trained Weight | Learning rate | Training epochs | Warm-up epochs |
|---|---|---|---|---|---|---|
| BraTS 2019 & BraTS 2020 | Swin-UNet | Swin-T | ImageNet-1k | 0.002 | 250 | 60 |
| | ViT+UPerNet | ViT-B/16 | ImageNet-21k | 0.002 | 250 | 25 |
| KiTS 2019 | Swin-UNet | Swin-T | ImageNet-1k | 0.002 | 500 | 20 |
| | ViT+UPerNet | ViT-B/16 | ImageNet-21k | 0.004 | 500 | 20 |

### A.3. Details about Generalization Capability Experiments.

We conduct ablation experiments to investigate the generalization capability of our Med-Tuning on the 3D baseline and pre-trained weight on the medical dataset. The Medical Segmentation Decathlon (MSD) (Antonelli et al., 2022) dataset includes 10 segmentation tasks covering various organs and image modalities. These tasks are intentionally diverse, presenting challenges like limited training data, class imbalances, multi-modality data, and small objects. In the main text, we have validated our approach on two MRI and one CT benchmark dataset. For ablation experiments, we selected one MRI and two CT datasets (i.e., Task02 Heart (MRI), Task06 Lung (CT), and Task09 Spleen (CT)) from the MSD dataset. Dataset pre-processing followed the protocol outlined in Swin UNETR (Tang et al., 2022). In Table 6, Memory(GB) represents memory usage during the fine-tuning,

Time(h) denotes the fine-tuning time, and Dice AVG signifies the average of multi-class Dice scores for the corresponding segmentation task.

### A.4. The position of the inserted parameters.

Regarding the insertion position for the parameters, for SwinUnet-Tiny and SwinUNETR, we have incorporated the Med-Adapter exclusively within each transformer layer of their encoder. This results in a total of 8 Med-Adapters, calculated from $4 \times 2$ (number of stages $\times$ number of layers in each stage). Within each stage, the second Med-Adapter is designated for Inter-FI. In the case of ViT+UPerNet, a Med-Adapter is inserted following every layer, amounting to a total of 12 (number of layers in ViT-B/16) Med-Adapters. Specifically, the Med-Adapters placed after the 2nd, 5th, 8th, and 11th layers are used for Inter-FI, maintaining a division of the ViT encoder into 4 stages, similar to the Swin Transformer setup. The relative positioning between Med-Adapters and Transformer blocks can be referenced in Figure 2 of our manuscript. Through our experiments, we have determined that the insertion position illustrated in Figure 2 of the manuscript represent the optimal configuration, as currently established.

### A.5. The scale of bottleneck features of Med-Adapter.

As indicated in Table 9 of our manuscript, the default Reduction Ratio is set to 6. Consequently, for all baselines, the scale of the bottleneck features of the adapter is represented by $L/6$, $L$ is the base scale of features in each stage (i.e., the scale of input features of Med-Adapter). Specifically, the scale of bottleneck features of 8 Med-Adapter in SwinUnet-Tiny or SwinUNETR are $[16, 16, 32, 32, 64, 64, 128, 128]$. The scale of bottleneck features of 12 Med-Adapter in ViT+UPerNet are both 128.

## Appendix B. Visualization Comparisons

### B.1. Visualization Comparisons with other PET method

Comparison with full fine-tuning, head tuning and previous PET methods in terms of the trade-off between the number of tuned parameters and segmentation accuracy is shown in Figure 3. The experiments were conducted using ViT+UPerNet as the baseline on the BraTS 2019 dataset. The horizontal axis represents the parameters involved in model training during the fine-tuning stage, while the vertical axis denotes the mean Dice scores for ET, WT, and TC. Our method achieves much better segmentation performance than full fine-tuning and previous state-of-the-art PET methods with much less tuned parameters.

### B.2. Visualization of BraTS 2019

Qualitative results of BraTS 2019 datasets are shown in Figure 4, with the comparison with full fine-tuning, ST-Adapter and VPT. As the labels for the validation set are not available, five-fold cross-validation is conducted on the training set for visualization. Our method recognizes brain tumors in enhancing and non-enhancing regions more accurately and reduces missed or false identification of the peritumoral edema in general.

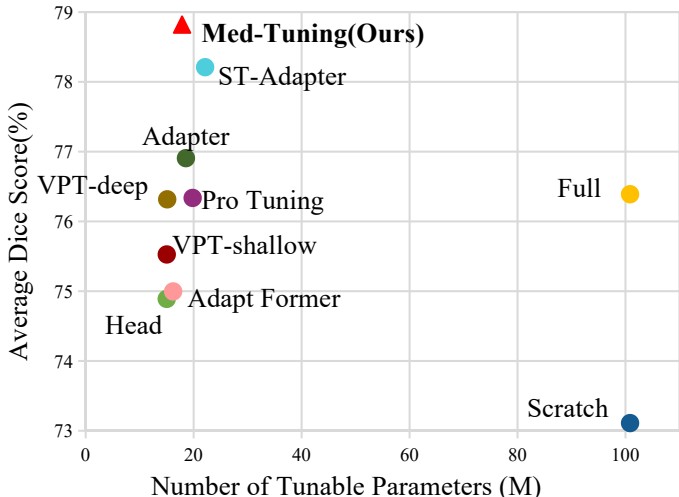

Figure 3: Comparison with previous PET methods in terms of the number of tuned parameters and Dice scores.

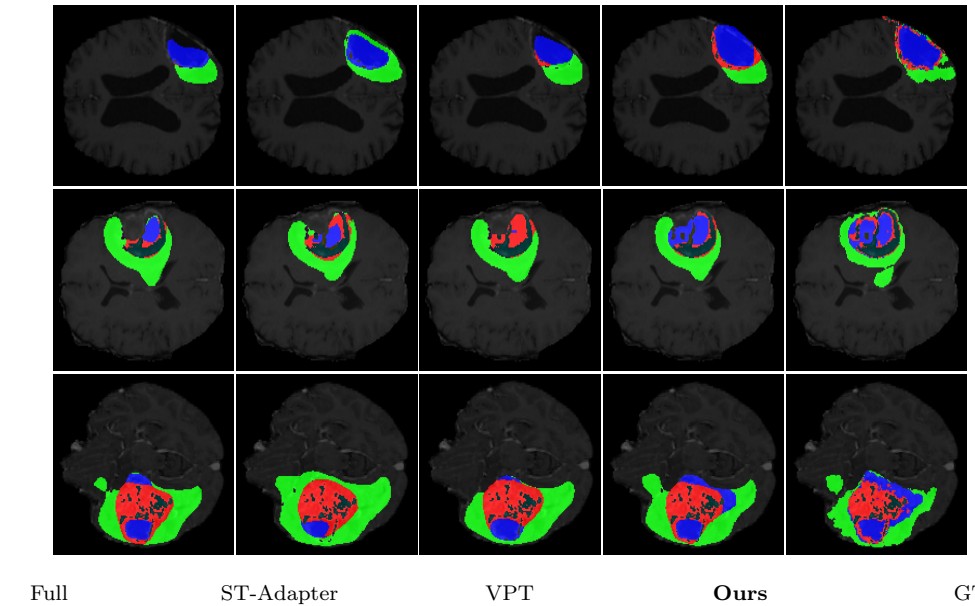

Figure 4: The visual comparison of segmentation results on BraTS 2019. The blue, red and green regions denote the enhancing tumors, non-enhancing tumors, and peritumoral edema. GT=Ground Truth.

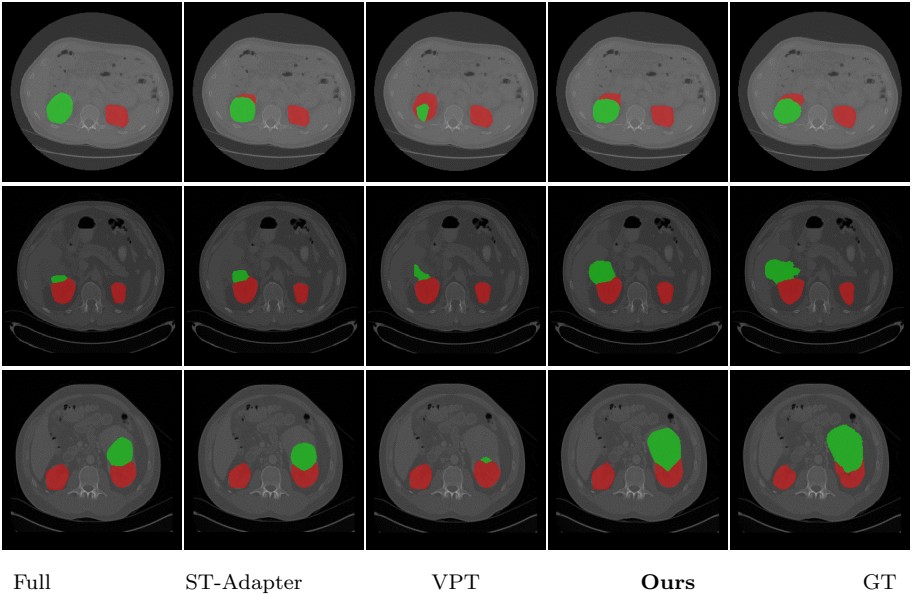

Full      ST-Adapter      VPT      **Ours**      GT

Figure 5: The visual comparison of segmentation results on KiTS 2019. The red and green regions denote the kidneys and kidney tumors. GT=Ground Truth.

### B.3. Visualization of KiTS 2019

As the labels for the validation set are not available, five-fold cross-validation is conducted on the training set for visualization. The qualitative results of KiTS 2019 datasets, depicted in Figure 5, highlight the superior performance of our method in organ and tumor segmentation compared to full fine-tuning, ST-Adapter, and VPT. Our approach demonstrates enhanced accuracy in segmenting organs and tumor types, producing finer-grained segmentation masks for corresponding tumors.

## Appendix C. More Ablation Studies and Analysis

### C.1. Reduction Ratio in Bottleneck Design.

| Method | Tuned Params(M) | Inserted Params(M) | Dice (%) ↑ | | | |
|---|---|---|---|---|---|---|
| | | | ET | WT | TC | Avg. |
| $\alpha=2$ | 10.064 | 3.313 | 76.89 | 90.14 | 81.92 | 82.99 |
| $\alpha=4$ | 7.994 | 1.243 | **77.22** | 90.09 | 81.59 | 82.97 |
| $\alpha=6$ | 7.489 | 0.738 | 77.06 | **90.28** | **82.71** | **83.35** |
| $\alpha=8$ | 7.271 | 0.520 | 76.94 | 89.62 | 80.74 | 82.44 |

Table 9: Ablation study on reduction ratio $\alpha$. Swin-UNet with Swin-T pre-trained on supervised ImageNet-1k.

We analyze the effect of different reduction ratios of the bottleneck structure in our Med-Adapter. Note that the reduction ratio $\alpha$ here is a key factor that influences the tuned parameters introduced by our Med-Adapter. Four diverse settings of $\alpha$ are selected. As shown in Table 9, Med-Tuning achieves a promising trade-off between segmentation accuracy and the tuned parameter costs with $\alpha = 6$. On this basis, higher $\alpha$ would cause inferior model performance because of the deteriorated representation capability with limited tuned parameters, while lower $\alpha$ would lead to a certain degree of information redundancy and a sharp increase of tuned parameters, resulting in both decreased segmentation accuracy and high training costs.

## C.2. Design for Global Dependency Modeling.

Table 10: Ablation study on different designs for global dependency modeling. The baseline is Swin-UNet with Swin-T pre-trained on supervised ImageNet-1k. DWConvK denotes depth-wise convolution with a kernel size of K × K.

| Method | Tuned Params(M) | Inserted Params(M) | Dice (%) ↑ | | | |
|---|---|---|---|---|---|---|
| | | | ET | WT | TC | Avg. |
| DWConv9 | 7.837 | 1.086 | 76.48 | **90.58** | 81.10 | 82.72 |
| DWConv11 | 8.126 | 1.375 | 76.82 | 89.40 | 80.05 | 82.09 |
| FFT | 7.994 | 1.243 | **77.22** | 90.09 | **81.59** | **82.97** |

In order to pursue the most effective and parameter-efficient architecture of our proposed Med-Adapter, we also investigate different designs for the global branch in our Med-Adapter block to achieve global dependency modeling. Since convolutional blocks with a large kernel size or self-attention are usually adopted by previous works for global contextual modeling and the baseline Swin-UNet itself consists of plenty of self-attention operation in each local window, we take the depth-wise convolution with a kernel size of 9 and 11 separately to replace our originally employed Fast Fourier Transform (i.e., FFT) branch for a comprehensive comparison. The comparison of the segmentation performance and tuned model parameters is shown in Table 10. It can be noticed that by taking advantage of the parameter-efficient FFT branch for effective long-range context modeling, the architecture with the FFT branch achieves the optimal trade-off between model performance and tuned parameters, reaching the best segmentation accuracy with only 1.243M introduced model parameters. In contrast, too large kernel size of the employed convolutions (i.e., DWConv11) will result in a burdensome model structure and a large amount of tuned parameter cost.

## C.3. Intra-FE Module Design.

We first probe into the rationale of the proposed Intra-FE Module without Inter-FI on Swin-UNet baseline. Swin-UNet with Swin-T pre-trained on supervised ImageNet-1k is taken as a baseline. As presented in Table 11, the introduction of MLB, FGB, or channel mixing consistently leads to a considerable performance increase. Specifically, with only 0.002M additional tuned parameters, the FGB branch greatly improves the segmentation

Table 11: Ablation study on Intra-FE. The first row is the result of the Vanilla Adapter.

| MLB | FGB | $Conv_{1\times1\times1}$ | Tuned Params(M) | Inserted Params(M) | Dice (%) ↑ | | |
|---|---|---|---|---|---|---|---|
| | | | | | ET | WT | TC |
| - | - | - | 7.541 | 0.790 | 75.13 | 87.50 | 75.29 |
| ✓ | - | - | 7.574 | 0.823 | 75.19 | 89.44 | 80.89 |
| ✓ | ✓ | - | 7.577 | 0.825 | 75.30 | 89.93 | **81.93** |
| ✓ | ✓ | ✓ | 7.675 | 0.924 | **77.10** | **90.05** | 81.02 |

Table 12: Ablation study on inter-FI.

| Method | Tuned Params(M) | Inserted Params(M) | Dice (%) ↑ | | |
|---|---|---|---|---|---|
| | | | ET | WT | TC |
| Add | 7.896 | 1.144 | 75.79 | 88.99 | 79.00 |
| Max | 7.896 | 1.144 | 75.22 | 89.72 | 81.41 |
| Concat | 7.994 | 1.243 | **77.22** | **90.09** | **81.59** |

accuracy, showing the effectiveness and parameter efficiency of our employed FGB branch. Additionally, channel mixing further boosts the performance by a large margin, especially on ET (↑ 1.80%).

### C.4. Inter-FI Module Design.

After investigating the effect of the intra-stage feature enhancement, we further verify the effectiveness of the inter-stage feature interaction, as shown in Table 12. Compared with the intra-only structure (i.e., without the feature connectivity between adjacent Med-Adapters), the model with inter-stage achieves a considerable performance gain with only 0.319M extra parameters for feature alignment among adjacent stages, showing the effectiveness of our inter-stage interaction. Unlike concatenation which maintains the feature representations of different stages as much as possible, direct addition or taking the maximum value (at each pixel) of neighboring feature maps with diverse semantic levels would unintentionally degrade the original feature representation, resulting in a sharp decrease in segmentation performance.

### C.5. Decoder Design.

Here we explore the effect of different decoder designs in our architecture. Although the backbone is frozen and only the inserted Med-Adapters as well as the decoder are updated during fine-tuning, the essentially tuned model parameters introduced by the segmentation decoder can not be reckoned as negligible. In other words, to pursue an extremely PET framework, the design of the employed decoder should be sufficiently lightweight with strictly controlled model parameters. Thus, various segmentation decoders with greatly varied model complexity are introduced respectively for a thorough analysis. As shown in Table 13, ViT-B/16 with the SETR-MLA decoder reaches the best trade-off between segmentation accuracy and tuned parameter costs, benefiting from the effective multi-scale

Table 13: Ablation study on decoder design. ViT-B/16 is pre-trained on supervised ImageNet-1k.

| Method | Tuned Params(M) | Decoder Params(M) | Dice (%) ↑ | | | |
|---|---|---|---|---|---|---|
| | | | ET | WT | TC | Avg. |
| UPerNet (Default) | 19.562 | 15.095 | 68.27 | 87.22 | 81.63 | 79.04 |
| U-Net | 9.269 | 4.712 | 67.68 | **88.08** | 81.72 | 79.16 |
| SETR-MLA | 8.347 | 3.790 | 68.12 | 87.91 | **81.98** | **79.34** |
| SETR-Naive | 5.004 | 0.447 | **69.11** | 86.93 | 81.71 | 79.25 |
| SETR-PUP | 5.200 | 0.643 | 68.55 | 86.51 | 80.42 | 78.49 |

feature aggregation. Besides, taking the simplest SETR-Naive that is composed of a convolution and an interpolation operation for upsampling as the decoder leads to the lowest tuned parameters 5.004M while achieving promising segmentation performance with an average Dice score of 79.34%. It can be seen from Table 13 that although the decoder size dominantly decides the overall tuned parameters, it does not show a direct impact on model performance.

## C.6. Data Efficiency.

Table 14: Ablation study on data efficiency property with pre-trained ViT-B/16.

| Dataset Ratio | Method | Memory Cost (GB)↓ | Training Time (h)↓ | Dice (%) ↑ | | | HF (mm)↓ | | |
|---|---|---|---|---|---|---|---|---|---|
| | | | | ET | WT | TC | ET | WT | TC |
| 100% | Full | 16.55 | 1.34 | 68.04 | 85.74 | 76.58 | 6.94 | 7.28 | 7.99 |
| 100% | Ours | 13.53 | 1.20 | **75.46** | **86.80** | **86.24** | **3.78** | 6.94 | **4.34** |
| 75% | Ours | 13.53 | 1.05 | 69.12 | 86.69 | 78.06 | 6.33 | **6.01** | 6.63 |
| 50% | Ours | 13.53 | 0.72 | 69.19 | 86.26 | 77.26 | 6.28 | 7.03 | 7.12 |
| 25% | Ours | 13.53 | 0.39 | 67.43 | 85.64 | 74.57 | 6.32 | 7.71 | 8.14 |
| 5% | Ours | 13.53 | 0.17 | 59.61 | 80.44 | 64.01 | 15.07 | 16.64 | 16.36 |

At last, we also explore the data efficiency property of our method by examining performance across various training data ratios, particularly in low-data settings. Table 14 shows the quantitative comparison with different numbers of training samples. Our Med-Tuning can already achieve comparable performance to full fine-tuning using only **25%** training data. As the scale of training data increases, our method consistently improves the segmentation accuracy, with reduced training time and memory cost compared with full fine-tuning.

## C.7. Other Weight Pre-trained on Medical Image Datasets.

Med-Tuning is not solely focused on pushing SOTA. Instead, it allows us to capitalize on the extensive progress made in natural image processing. This perspective underscores our

belief in the potential and value of integrating advancements from one domain to enhance the capabilities and applications in another.

Indeed, as highlighted in recent literature (Liu et al., 2023b; Silva-Rodríguez et al., 2023; Ulrich et al., 2023), there have been significant advancements in the field of medical image pre-trained models. Nevertheless, due to the considerable constraints of time, monetary resources, and clinical applicability faced by many researchers working on medical image pre-training, the pace of updates and the scale of medical image pre-training efforts still trail behind those in the natural image domain. Additionally, the use of many open-source codes in the medical imaging field presents a high threshold. Therefore, the vast array of convenient and accessible large-scale pre-trained weights from the natural image domain have become our primary choice.

Based on the above choices, we hypothesize that: If Med-Tuning can tackle the more challenging task of a large domain shift from features pre-trained on natural 2D images to CT/MRI volumes, then it is also capable of addressing the comparatively easier task of domain shift from features pre-trained on medical images to CT/MRI volumes. The experimental results in our manuscript have demonstrated the feasibility of a broader transfer process, thereby validating the effectiveness of our proposed approach in achieving the former scenario.

Regarding the latter scenario, we conducted experiments using the same baselines and pre-trained weights as those in (Liu et al., 2023b; Silva-Rodríguez et al., 2023), following our default training setting. The comprehensive results of all experiments are depicted in Table 15.

Table 15: The comparison between original SwinUNETR, Universal Model and our proposed Med-Tuning. The performance is evaluated by average Dice scores. "W1" signifies the use of the model and pre-training weights from (Tang et al., 2022), while "W2" references the model and pre-training weights from (Liu et al., 2023b). "SCR" denotes the model is trained from scratch and "FULL" denotes the full fine-tuning mothod. The first two columns of scores were directly copied from (Liu et al., 2023b). The last four columns of scores were obtained through training using our framework.

| Dataset | SwinUNETR (SCR) | Universal Model (FULL) | Ours (SCR) | Ours (FULL, W1) | Ours (Med-Tuning, W1) | Ours (FULL, W2) | Ours (Med-Tuning, W2) |
|---|---|---|---|---|---|---|---|
| Task06 Heart | 68.90 | 67.15 | 65.82 | 67.69 | 78.09 | 68.37 | **78.53** |
| Task09 Heart | 95.80 | 96.71 | 95.76 | 96.52 | 97.06 | 96.35 | **97.60** |

From our results, Med-Tuning proves to be capable of consistently improving the precision in medical volumetric segmentation tasks by using medical pre-trained weights, requiring only a small number of training parameters for this enhancement. Besides, the trends observed in our experimental results suggest that our proposed approach can keep pace

Table 16: Comparisons of training time (hours) on BraTS2019 with SwinUnet and ViT+UPerNet backbone.

| Method | ViT+UPerNet | SwinUnet |
|---|---|---|
| Scratch | 1.74h | 1.26h |
| Full | 1.73h | 1.26h |
| Head | 1.28h | 1.02h |
| VPT-Shallow | 1.09h | 0.98h |
| VPT-Deep | 1.18h | 1.01h |
| Adapter | 1.77h | 1.30h |
| AdaptFormer | 1.44h | 1.18h |
| Pro-tuning | 1.84h | 1.47h |
| ST-Adapter | 1.79h | 1.55h |
| Ours | 1.88h | 1.51h |

with the development of visual models pre-trained in medical domain, aligning with the conclusions drawn at the end of our manuscript.

Finally, we would like to add that as demonstrated in our results shown in Table 10, using the W2 weights improved the Dice score by 0.6 over the W1 weights. Hence, we also look forward to the widespread development of large-scale pre-trained models like (Liu et al., 2023b) in medical domain and are excited about the potential to further enhance their performance using our Med-Tuning.

### C.8. Training Time.

Under default training settings, the training time of each method are listed in Table 16. The results indicate that the introduction of few new training parameters inevitably results in a slight increase in training. However, we achieves a commendable balance between training time cost and the tuning of parameters.

