# OpenReview forum: "Med-Tuning: A New Parameter-Efficient Tuning Framework for Medical Volumetric Segmentation"
_MIDL.io/2024/Conference — MIDL 2024 Poster_

### Official Review · Reviewer_H82q · 2024-02-20

**Confidence:** 3
**Preliminary Rating:** 3
**Recommendation:** Poster
**Final Rating:** 2

**Summary:**

This paper proposes to adapt backbones pre-trained on natural image classification, segmentation, or vision-language pre-training, for volumetric medical image segmentation. To do so, their Med-Tuning method includes special modules for transferring 2d architectures to 3d data, using adapters with 3D convolutions. Also, Med-Tuning includes inter-level feature concatenation, and FFT transforms. Med-Tuning is validated across several tasks using CT and MRI volumes, using different backbones.

**Strengths:**

- The problem tackled is important.
- The authors provide many experiments that properly validate each component of the proposed adapter.
- The authors compare their adapter with a relevant body of literature.
- Results demonstrate efficient and robust adaptation, with outstanding improvements over popular approaches such as finetuning and decoder training.

**Weaknesses:**

- The core idea of parameter-efficient fine-tuning implies re-using rich pre-trained features for segmentation and adapting them to downstream tasks by modifying a small set of new or existing parameters. Under this scope, using features pre-trained on natural images, and 2D data, proposes a large domain shift to the target domain: CT/MRI volumes. In addition, some dedicated designs, such as the proposed multi-scale local branch, are necessary to adapt the 2D model to 3D data. Thus, adapting such models, instead of recently released foundation models for 3D data (see [a,b,c]), might be unnecessary and suboptimal.
- Following the previous comment, there is one experiment using Swin UNETR, but this seems pre-trained only on 1 dataset, in contrast to [a,b,c]. Also, there is no direct comparison with natural image pre-training.
- Although the number of parameters is indicated for all PEFT methods in Table 1, this is not the case for training times. For some settings, training times might be larger even when using a small set of parameters, due to convergence speeds.
- Following the previous comment, some training details are missing: are the authors using a validation set? Are authors doing early stopping? Some methods might converge faster than others.
- Implementation details for baselines. There are some missing implementation details, mostly for the baseline methods. This might question if all baseline methods were properly adjusted (i.e. learning rate, the position of the inserted  parameters, scale of bottleneck features of the adapter, etc.).

[a] CLIP-Driven Universal Model for Organ Segmentation and Tumor Detection, ICCV, 2023.

[b] Towards Foundation Models and Few-Shot Parameter-Efficient Fine-Tuning for Volumetric Organ Segmentation, MICCAI MedAGI Workshop, 2023.

[c] MultiTalent: A Multi-Dataset Approach to Medical Image Segmentation, MICCAI, 2023.

**Detailed Comments:**

No additional comments.

**Justification Of Final Rating:**

I thank the authors for their detailed responses. However, after carefully reading the comments of other reviewers and the authors' responses, I believe that some important concerns have not been adequately addressed, which motivates me to lower my score. I include below the detailed weaknesses that motivate my decision.

1. **Experimental Setting and Baselines**. Focusing on the large-data regime, i.e. several hundred volumes for training, as stated by Reviewer ufgr, dataset-specific models (instead of adapted ones) might be an appealing solution, using dataset-specific architectures such as nnUnet, in particular its 3D version. These solutions are not considered in the manuscript. In addition, Scratch or Full training baselines (Table 1, 2, and 3) only receive 2D information, thus being a weakly implemented baseline compared to the 3D-based Adapter.


2. Med-Tuning framework introduces several **hyperparameters and modules**, evaluated in several ablation experiments (Table 4, Table 9, Table 10, Table 11, Table 12). Each of such decisions might vary (1%-2%) the obtained results. There are some tasks (e.g. BraTS) for which performance improvements are minor, which might question the actual performance gains to previous adapters if going through such empirical refinement.


3. **Med-Adapter configuration**. As questioned by Reviewer XA6b, the proposed adapter fails to integrate a motivated link for the addition of several components (e.g. FGB and Inter-FI), which are unproperly linked to previous literature adapters. In addition, some branches, such as inter-FI, introduce minor performance gains (Table 12).


4. **Comparison with 3D medical pre-trained foundation models**. The authors include in Table 6 a SwinUNETR model pre-trained uniquely on 1 dataset. Initial rebuttal experiments (Table 10 at Reviewer H82q discussion) suggest already an improvement when using large-scale pre-trained medical volumetric models with the proposed Med-Tuning adapter compared to Table 6 experiments. It is worth mentioning that such experiments involve fine-tuning the pre-trained decoder, which might distort rich pre-trained specialized features [e]. Such domain-specific models might require less tuning than natural-image pre-trained models, and thus, I do not agree that “by conducting comparisons within each pre-trained domain separately, we can adequately demonstrate the superiority of Med-Tuning”. As pointed out in my initial review, there exists a growing appearance of medical pre-trained volumetric models [a,b,c,d], some of them exploring parameter-efficient alternatives that benefit from the pre-trained decoder (e.g. Linear Probe then Fine-Tune in [c] or spatial adapters in [b]). Thus, the transferability of natural-image 2d pre-trained models for transferability to 3D medical image segmentation is not properly justified.

[a] CLIP-Driven Universal Model for Organ Segmentation and Tumor Detection, ICCV, 2023.

[b] Towards Foundation Models and Few-Shot Parameter-Efficient Fine-Tuning for Volumetric Organ Segmentation, MICCAI MedAGI Workshop, 2023.

[c] MultiTalent: A Multi-Dataset Approach to Medical Image Segmentation, MICCAI, 2023.

[d] UniSeg: A Prompt-driven Universal Segmentation Model as well as A Strong Representation Learning, MICCAI, 2023.

[e] Fine-Tuning can Distort Pretrained Features and Underperform Out-of-Distribution, ICLR, 2022.

**Justification Of The Preliminary Rating:**

The methodological description of the paper is sound, and the authors provide many experiments and adequate ablation experiments. Nevertheless, I have some concerns about the efficiency of adapting natural image pre-trained models and experimental designs for baseline methods.

**Questions To Address In The Rebuttal:**

Please, see Weaknesses.

**Special Issue:**

No

---

> ### Author Response · Authors · 2024-03-16
> **Sincerely appreciation for novelty, contribution, and the positive feedback**
>
> Thanks very much for acknowledging the novelty and contributions of our paper, as well as the provided positive feedback "The methodological description of the paper is sound, and the authors provide many experiments and adequate ablation experiments." on our extensive evaluations and robustness.
>
> We have addressed all your questions in detail in the Response section below. It is our sincere hope that our thorough explanations, coupled with additional experiments conducted to the best of our abilities, will contribute to an improved evaluation score of our work.
>
>
> **Our Responses to Paper Weaknesses:**
>
> > [**Q1**]. The core idea of parameter-efficient fine-tuning implies re-using rich pre-trained features for segmentation and adapting them to downstream tasks by modifying a small set of new or existing parameters. Under this scope, using features pre-trained on natural images, and 2D data, proposes a large domain shift to the target domain: CT/MRI volumes. In addition, some dedicated designs, such as the proposed multi-scale local branch, are necessary to adapt the 2D model to 3D data. Thus, adapting such models, instead of recently released foundation models for 3D data (see [a,b,c]), might be unnecessary and suboptimal.Following the previous comment, there is one experiment using Swin UNETR, but this seems pre-trained only on 1 dataset, in contrast to [a,b,c]. & Nevertheless, I have some concerns about the efficiency of adapting natural image pre-trained models.
>
> [**A1**].
>
> Thanks for this feedback. We appreciate the opportunity to clarify this point and remain open to further discussion that might arise regarding our work.
>
> **Firstly, we would like to assert that while our proposed method may be considered 'suboptimal,' it is certainly not 'unnecessary.'**
>
> Med-Tuning is not solely focused on pushing SOTA. Instead, it allows us to capitalize on the extensive progress made in natural image processing. This perspective underscores our belief in the potential and value of integrating advancements from one domain to enhance the capabilities and applications in another.
>
> Indeed, as highlighted in recent literature [a][b][c], there have been significant advancements in the field of medical image pre-trained models. Nevertheless, due to the considerable constraints of time, monetary resources, and clinical applicability faced by many researchers working on medical image pre-training, the pace of updates and the scale of medical image pre-training efforts still trail behind those in the natural image domain. Additionally, the use of many open-source codes in the medical imaging field presents a high threshold. Therefore, the vast array of convenient and accessible large-scale pre-trained weights from the natural image domain have become our primary choice.
>
> Based on the above choices, we hypothesize that: If Med-Tuning can tackle the more challenging task of *a large domain shift from features pre-trained on natural 2D images to CT/MRI volumes*, then it is also capable of addressing the comparatively easier task of *domain shift from features pre-trained on medical images to CT/MRI volumes*. The experimental results in our manuscript have demonstrated the feasibility of a broader transfer process, thereby validating the effectiveness of our proposed approach in achieving the former scenario.
>
> Regarding the latter scenario, we conducted experiments using the same baselines and pre-trained weights as those in [a][b], following our default training setting. The comprehensive results of all experiments are depicted in Table 10.

---

> ### Author Response · Authors · 2024-03-16
>
> **Table 10: The 5-fold cross-validation performance on MSD. These are the comparison between original SwinUNETR, Universal Model and our proposed Med-Tuning. The performance is evaluated by average Dice scores. "W1" signifies the use of the model and pre-training weights from [d], while "W2" references the model and pre-training weights from [a]. "SCR" denotes the model is trained from scratch and "FULL" denotes the full fine-tuning mothod. The first two columns of scores were directly copied from [a]. The last four columns of scores were obtained through training using our framework.**
> | Dataset | SwinUNETR (SCR) | Universal Model (FULL) | Ours (SCR) | Ours (FULL, W1) | Ours (Med-Tuning, W1) | Ours (FULL, W2) | Ours (Med-Tuning, W2) |
> |:------:|:-------:|:-------:|:-------:|:-------:|:-------:|:-------:|:-------:|
> |Task06 Heart| 68.90 |67.15 | 65.82 | 67.69 | 78.09 | 68.37 | 78.53 |
> |Task09 Heart| 95.80 |96.71 | 95.76 | 96.52 | 97.06 | 96.35 | 97.60 |
>
> From our results, Med-Tuning proves to be capable of consistently improving the precision in medical volumetric segmentation tasks by using medical pre-trained weights, requiring only a small number of training parameters for this enhancement. Besides, the trends observed in our experimental results suggest that our proposed approach can keep pace with the development of visual models pre-trained in medical domain, aligning with the conclusions drawn at the end of our manuscript.
>
> Finally, we would like to add that as demonstrated in our results shown in Table 10, using the W2 weights improved the Dice score by 0.6 over the W1 weights. Hence, we also look forward to the widespread development of large-scale pre-trained models like [a] in medical domain and are excited about the potential to further enhance their performance using our Med-Tuning.
>
> [a] CLIP-Driven Universal Model for Organ Segmentation and Tumor Detection, ICCV, 2023.
>
> [b] Towards Foundation Models and Few-Shot Parameter-Efficient Fine-Tuning for Volumetric Organ Segmentation, MICCAI MedAGI Workshop, 2023.
>
> [c] MultiTalent: A Multi-Dataset Approach to Medical Image Segmentation, MICCAI, 2023.
>
> [d] Self-Supervised Pre-Training of Swin Transformers for 3D Medical Image Analysis. CVPR, 2021.
>
>
> > [**Q2**]. Also, there is no direct comparison with natural image pre-training.
>
> [**A2**]. By conducting comparisons within each pre-trained domain separately, we can adequately demonstrate the superiority of Med-Tuning. Thus, it is not essential to use the same baseline for a comparative analysis of Med-Tuning across both pre-trained domains simultaneously. Therefore, we believe that the existing experiments already provide convincing evidence to support our claims. Besides, it appears that among the current open-source pre-trained models, there are few that possess both natural image pre-training and medical image pre-training weight. We sincerely hope for your understanding and welcome any further discussions or suggestions regarding our work.

---

> > ### Comment · Reviewer_H82q · 2024-03-23
> >
> > Thank you for the detailed responses to my comments. However, I have one last question that I would kindly ask the authors to clarify:
> >
> > Regarding the experiments using SwinUNETR pre-trained on medical datasets (Table 6 of the paper and new experiments in rebuttal Table 10 using UniversalModel). How is Med-Tuning implemented in this case? SwinUNETR already has a pre-trained encoder-decoder architecture for volumetric data. For example, the methodological description of Med-Tuning (Section 3.2 and Figure 2) involves training a decoder from scratch.

---

> > > ### Author Response · Authors · 2024-03-24
> > >
> > > Many thanks for your kind comments and great interests on our work. Please let me clearly clarify this last question in the followings.
> > >
> > > **1. Decoder training**
> > >
> > > As indicated in the methodological section 3.2 and Figure 2 of our paper, Med-Tuning incorporates a pre-trained backbone along with a segmentation decoder that does not inherently require pre-trained weights for its operation.
> > >
> > > To provide further detail:
> > > - On one hand, in cases where a backbone such as ViT consists of only an encoder, it is necessary to integrate an effective decoder to address our segmentation tasks. In such scenarios, training the decoder from scratch becomes inevitable.
> > > - On the other hand, when a backbone already has a pre-trained encoder-decoder architecture, like SwinUnet and SwinUNETR, we directly utilize its weights to initialize the parameters for both the encoder and decoder. In these instances, the decoder can be trained using a full fine-tuning approach. Specifically for SwinUnet, we adhere to the method detailed in its original paper for symmetrically initializing the decoder with the pre-trained weights of Swin Transformer.
> > >
> > > **2. Med-Adapter implementation details**
> > >
> > > In reference to the nuanced implementation details of the Med-Adapter, we wish to provide clarification regarding the adjustments made to accommodate the SwinUNETR architecture, which serves as a 3D baseline.
> > > Specifically, the downsample and upsample linear layers have been adapted to utilize downsample and upsample 3D convolutions with a kernel size of 1.
> > > Moreover, the reshape operations traditionally require before and after the Depth-Wise Convolution, denoted as $R_{volume}$ and $R_{flatten}$ in equations 5 and 7 of our manuscript, have been found unnecessary for SwinUNETR and thus have been omitted.
> > >
> > > These modifications constitute the only deviations from our Med-Adapter’s standard implementation.
> > >
> > > We hope these detailed explanations above illuminate the bespoke adaptations implemented for the SwinUNETR architecture in our study, giving us a precious opportunity to strive for your raised final rating.

---

> ### Author Response · Authors · 2024-03-16
>
> > [**Q3**]. Although the number of parameters is indicated for all PEFT methods in Table 1, this is not the case for training times. For some settings, training times might be larger even when using a small set of parameters, due to convergence speeds.
>
> [**A3**]. Under default training settings, the training time of each method are listed in Table 11. The results indicate that the introduction of few new training parameters inevitably results in a slight increase in training. However, we achieves a commendable balance between training time cost and the tuning of parameters.
>
> **Table 11: Comparisons of training time (hours) on BraTS2019 with SwinUnet and ViT+UPerNet backbone.**
> | Method | ViT+UPerNet | SwinUnet |
> |:------|:-------:|:-------:|
> | Scratch | 1.74 | 	1.26
> | Full | 1.73 | 1.26
> | Head | 1.28 | 1.02
> | VPT-Shallow | 1.09 | 0.98
> | VPT-Deep | 1.18 | 1.01
> | Adapter | 1.77 | 1.30
> | AdaptFormer | 1.44 | 1.18
> | Pro-tuning | 1.84 | 1.47
> | ST-Adapter | 1.79 | 1.55
> | Ours | 1.88 | 1.51
>
>
> > [**Q4**]. Following the previous comment, some training details are missing: are the authors using a validation set? Are authors doing early stopping? Some methods might converge faster than others.
>
> [**A4**]. Due to the relatively small size of medical imaging datasets, it is vital to utilize the entire official training data. Therefore, we trained our models in the following manner:
>
> Firstly, we initially employed 3-fold cross-validation, dividing the entire official training set into three parts. Each part was alternately used as the validation set, with the remaining two parts serving as the training set, thereby creating three sets of train-validation pairs.
>
> With each change in hyperparameters, we conducted training and validation across these three train-validation pairs, ultimately calculating their average performance as the representation of effectiveness on the train-validation pairs. Through iterative adjustments of the hyperparameters, we identified a set of hyperparameters that exhibited optimal average performance across the train-validation pairs.
>
> Subsequently, employing this optimal set of hyperparameters, we utilized all data from the official training set for training, followed by testing on the official test set and submitting these results to the official website for scoring. This experiment was replicated three times, and the mean of these repetitions was used to determine the scores presented in Tables 1, 2, and 3 of our manuscript.

---

> ### Author Response · Authors · 2024-03-16
>
> > [**Q5**]. Implementation details for baselines. There are some missing implementation details, mostly for the baseline methods. This might question if all baseline methods were properly adjusted (i.e. learning rate, the position of the inserted parameters, scale of bottleneck features of the adapter, etc.). & Nevertheless, I have some concerns about the efficiency of ... and experimental designs for baseline methods.
>
> [**A5**].
>
> **(1) Implementation details for 3 baselines.** Below, we list some of the config used for the SwinUnet and ViT+UPerNet baselines in our manuscript. The configs of SwinUNETR is consistent with [1].
> ```Python
> # SwinUnet
> _C.MODEL.SWIN.PATCH_SIZE = 4
> _C.MODEL.SWIN.EMBED_DIM = 96
> _C.MODEL.SWIN.DEPTHS = [2, 2, 2, 2]
> _C.MODEL.SWIN.DEPTHS_DECODER = [2, 2, 2, 1]
> _C.MODEL.SWIN.NUM_HEADS = [3, 6, 12, 24]
> _C.MODEL.SWIN.WINDOW_SIZE = 4
> _C.MODEL.SWIN.MLP_RATIO = 4.
> _C.MODEL.SWIN.PRE_CKPT='../../ckpt/swin_tiny_patch4_window7_224.pth' # supervised pre-training
>
> # ViT+UPerNet
> _C.MODEL.VIT.PATCH_SIZE=16
> _C.MODEL.VIT.EMBED_DIM=768
> _C.MODEL.VIT.DEPTH=12
> _C.MODEL.VIT.NUM_HEADS=12
> _C.MODEL.VIT.PRE_CKPT_SUP='../../ckpt/ViT-B_16.npz' # supervised pre-training
> ```
>
> **(2) Learning rate:** We have identified the optimal hyperparameter settings for different baselines through extensive experiments. For the experiments on the BraTS 2019 & BraTS 2020 datasets, the learning rate is set at 0.002 for SwinUnet and 0.002 for ViT+UPerNet. For the experiments on the KiTS 2019 dataset, the learning rate is 0.002 for SwinUnet and 0.004 for ViT+UPerNet. Additional implementation details, such as training epochs and warm-up periods, are provided in Table 12.
>
> **Table 12: Implementation details.**
> | Dataset | Baseline | Encoder Backbone | Learning Rate| Training Epochs| Warm-up Epochs|
> |:------:|:-------:|:-------:|:-------:|:-------:|:-------:|
> |BraTS 2019 & BraTS 2020|SwinUnet|Swin-Tiny|0.002|250|60|
> |BraTS 2019 & BraTS 2020|ViT+UPerNet|ViT-B/16|0.002|250|25|
> |KiTS 2019|SwinUnet|Swin-Tiny|0.002|500|20|
> |KiTS 2019|ViT+UPerNet|ViT-B/16|0.004|500|20|
> |MSD Task02 Heart|SwinUNETR|Swin-Tiny|0.005|400|40|
> |MSD Task06 Lung|SwinUNETR|Swin-Tiny|0.005|400|50|
> |MSD Task09 Spleen|SwinUNETR|Swin-Tiny|0.005|400|40|
>
> **(3) The position of the inserted parameters:** Regarding the insertion position for the parameters, for SwinUnet-Tiny and SwinUNETR, we have incorporated the Med-Adapter exclusively within each transformer layer of their encoder. This results in a total of 8 Med-Adapters, calculated from $4\text{ (number of stages)}×2\text{ (number of layers in each stage)}$. Within each stage, the second Med-Adapter is designated for Inter-FI. In the case of ViT+UPerNet, a Med-Adapter is inserted following every layer, amounting to a total of $12\text{ (number of layers in ViT-B/16)}$ Med-Adapters. Specifically, the Med-Adapters placed after the $2nd, 5th, 8th,$ and $11th$ layers are used for Inter-FI, maintaining a division of the ViT encoder into 4 stages, similar to the Swin Transformer setup. The relative positioning between Med-Adapters and Transformer blocks can be referenced in Figure 2 of our manuscript. Through our experiments, we have determined that the insertion position illustrated in Figure 2 of the manuscript represent the optimal configuration, as currently established.
>
> **(4) The scale of bottleneck features of Med-Adapter:** As indicated in Table 9 of our manuscript, the default Reduction Ratio is set to 6. Consequently, for all baselines, the scale of the bottleneck features of the adapter is represented by $L/6$, $L$ is the base scale of features in each stage (i.e., the scale of input features of Med-Adapter). Specifically, the scale of bottleneck features of 8 Med-Adapter in SwinUnet-Tiny or SwinUNETR are $[16, 16, 32, 32, 64, 64, 128, 128]$. The scale of bottleneck features of 12 Med-Adapter in ViT+UPerNet are both $128$.
>
> [1] Tang, Y., Yang, D., Li, W., Roth, H.R., Landman, B., Xu, D., Nath, V., Hatamizadeh, A.: Self-supervised pre-training of swin transformers for 3D medical image analysis. In: CVPR (2021)

---

### Official Review · Reviewer_XA6b · 2024-03-01

**Confidence:** 4
**Preliminary Rating:** 3
**Recommendation:** Poster
**Final Rating:** 3.5

**Summary:**

This manuscript present the MedTuning framework that proposes a way to fine-tune foundational models pre-trained on 2D images for medical imaging segmentation using a Fourier transform based adapter inserted in the decoding part of the network at a lesser cost in terms of memory and tuned parameters
Experiments are run in comparison to other fine-tuning methods over the KiTS and BRATS datasets using two different baseline architectures, namely SwinUNet and ViT + UPerNet.

**Strengths:**

- Extensive experiments with multiple datasets and baseline architectures
- Large assessment of other possible methods over these datasets
- Clear identification of the current gaps in the literature and the problem to solve

**Weaknesses:**

- The overall clarity of the paper is quite poor and it becomes difficult to follow the rationale behind the methods.
- The link between the equations and the text is sometimes unclear. For instance, when reading the description of the Adapter, a natural translation would be $Adapter(X) = W_{up}(\sigma(W_{down}(X)))$. It is quite unclear where the addition term X+ comes from from the textual description
- There is no indication of results variability and no adequate statistical testing which may lead to overstatements of the results.

**Detailed Comments:**

The authors have clearly put a lot of effort in preparing the experiments and designing their solution but the numerous language issues throughout the papers make it very difficult to follow.
There are many nominal statements without verbs (e.g : Secondly, improve the flexibility and lightness of the framework on p4)
Some of the sentences are very convoluted and missing some words
References to future results (section 4.2 referred to in section 3.2) does not help with clarity.

**Justification Of Final Rating:**

The authors have interestingly addressed most of the comments and have put some effort in clarifying their statements and their claims. Given the possible interest of the community in the proposed solution, I would be happy to move my score to a borderline accept

**Justification Of The Preliminary Rating:**

The idea is quite interesting with comprehensive experiments but the overall clarity is poor and the conclusions may overstate the results. Statistical analysis is fully lacking and there is no clear link between the different pieces of the framework in terms of justification and/or explanation

**Questions To Address In The Rebuttal:**

Please re-ascertain the strengths of the results given adequate statistical analysis of the performance comparisons and temper the conclusions accordingly.

**Special Issue:**

No

---

> ### Author Response · Authors · 2024-03-16
> **Sincerely appreciation for your recognition of the novelty, contributions, extensive experiments and clear identification of the current gaps**
>
> We sincerely appreciate your recognition of the novelty and contributions of our paper, as well as the provided positive comments on our extensive experiments and clear identification of the current gaps. We have addressed all of your questions in detail in the following Response section and will incorporate all feedback in the final version. We genuinely hope that our detailed explanations and the additionally supplemented experiments with our best efforts will give us the precious opportunity to raise the evaluation score of our work in your perspective.
>
> **Our Responses to Paper Weaknesses:**
>
> > [**Q1**]. The overall clarity of the paper is quite poor and it becomes difficult to follow the rationale behind the methods.
>
> [**A1**]. Following your valuable suggestions, along with feedback obtained from several additional scholars invited to read this work, we have enhanced the overall clarity of the paper, making it easier to follow and understand. The corresponding revisions will be incorporated into the final version of the paper upon acceptance.
>
>
>
>
> > [**Q2**]. The link between the equations and the text is sometimes unclear. For instance, when reading the description of the Adapter, a natural translation would be $\text{Adapter}(X) = W_{up}(\sigma(W_{down}(X)))$. It is quite unclear where the addition term X+ comes from from the textual description.
>
> [**A2**]. Thanks for your detailed review. According to the definition of the Adapter in [1], the textual description in the paper should be based on formula(1): $\text{Adapter}(X) = X + W_{up}(\sigma(W_{down}(X)))$. A vanilla Adapter contains a skip-connection, allowing the adapter module to approximate an identity function when the parameters of the projection layers are close to zero.
>
> [1] Parameter-efficient transfer learning for NLP. ICML, 2019.

---

> ### Author Response · Authors · 2024-03-16
>
> > [**Q3**]. There is no indication of results variability and no adequate statistical testing which may lead to overstatements of the results. & Please re-ascertain the strengths of the results given adequate statistical analysis of the performance comparisons and temper the conclusions accordingly.
>
> [**A3**]. Per your advice, we supplement some representative raw results accompanied by their standard deviations for statistical analysis, which are listed below in Table 6-9 for your convenience to check. The results indicate that, despite **minor overlaps in certain cases**, our method is **generally comparable to or surpasses** the results of full fine-tuning and most of the previous PET methods.
>
> **Table 6: Performance comparison on KiTS 2019 with SwinUnet backbone.**
> | Method | Kidney Dice | Tumor Dice |
> |:------|:-------:|:-------:|
> | Scratch | 94.33±0.06 | 61.10±0.40 |
> | Full | 94.68±0.07 | 62.13±0.97 |
> | Head | 91.95±0.03 | 53.93±0.35 |
> | VPT-Shallow | 91.72±0.09 | 54.86±0.71 |
> | VPT-Deep | 91.53±0.12 | 53.41±0.76 |
> | Adapter | 93.02±0.04 | 57.15±0.62 |
> | AdaptFormer | 93.74±0.03 | 59.79±0.98 |
> | Pro-tuning | 90.34±0.08 | 51.19±0.46 |
> | ST-Adapter | 92.97±0.07 | 57.33±0.53 |
> | Ours | 95.69±0.05 | 70.14±0.54 |
>
> **Table 7: Performance comparison on BraTS2019 with SwinUnet backbone.**
> | Method | Dice(ET) | Dice(WT) | Dice(TC) | Hausdorff(ET)| Hausdorff(WT)| Hausdorff(TC)|
> |:------|:-------:|:-------:|:-------:|:-------:|:-------:|:-------:|
> Scratch|78.38±0.29|88.59±0.12|76.46±0.45|6.055±0.202|10.651±0.531|9.176±0.301|
> Full|78.26±0.25|89.56±0.07|79.16±0.63|4.327±0.598|6.149±0.179|6.704±0.120|
> Head|78.07±0.07|88.68±0.09|77.26±0.33|5.021±0.374|6.697±0.082|7.091±0.392|
> VPT-Shallow|77.16±0.18|88.30±0.15|76.77±0.27|5.421±0.108|6.154±0.044|7.345±0.053|
> VPT-Deep|77.02±0.22|88.65±0.07|76.91±0.32|5.297±0.220|7.088±0.157|7.940±0.132|
> Adapter|77.98±0.25|89.22±0.19|78.02±0.46|5.298±0.061|6.622±0.094|8.490±0.359|
> AdaptFormer|77.69±0.13|88.61±0.14|76.83±0.68|4.909±0.802|6.290±0.058|7.885±0.287|
> Pro-tuning|78.58±0.14|89.33±0.04|78.79±0.36|5.273±0.040|6.410±0.323|8.236±0.376|
> ST-Adapter|78.40±0.19|89.54±0.02|77.44±0.11|4.751±0.068|6.013±0.037|7.405±0.317|
> Ours|78.51±0.16|89.68±0.03|80.44±0.12|4.003±0.062|5.517±0.043|5.756±0.304|
>
> **Table 8: Performance comparison on BraTS2019 with ViT+UPerNet backbone.**
> | Method | Dice(ET) | Dice(WT) | Dice(TC) | Hausdorff(ET)| Hausdorff(WT)| Hausdorff(TC)|
> |:------|:-------:|:-------:|:-------:|:-------:|:-------:|:-------:|
> | Scratch | 64.96±0.30 | 83.03±0.17 | 71.34±0.39 | 7.635±0.162 | 10.602±0.504 | 10.942±0.186 |
> | Full | 68.49±0.20 | 85.56±0.05 | 75.12±0.56 | 6.672±1.492 | 7.878±0.155 | 10.525±1.091 |
> | Head | 65.71±0.04 | 84.19±0.08 | 74.77±0.29 | 6.128±0.298 | 7.505±0.036 | 7.864±0.324 |
> | VPT-Shallow | 66.02±0.11 | 84.72±0.14 | 75.84±0.28 | 6.114±0.087 | 7.506±0.020 | 8.471±0.014 |
> | VPT-Deep | 67.01±0.22 | 85.14±0.04 | 76.80±0.38 | 6.064±0.174 | 7.717±0.134 | 7.648±0.094 |
> | Adapter | 68.30±0.12 | 85.37±0.15 | 77.05±0.42 | 5.501±0.031 | 7.636±0.053 | 7.986±0.305 |
> | AdaptFormer | 65.88±0.16 | 84.34±0.14 | 74.77±0.62 | 6.652±0.568 | 8.204±0.030 | 8.430±0.232 |
> | Pro-tuning | 67.18±0.16 | 85.32±0.04 | 76.51±0.24 | 5.805±0.014 | 7.073±0.266 | 7.564±0.262 |
> | ST-Adapter | 69.18±0.19 | 86.27±0.04 | 79.18±0.07 | 6.077±0.043 | 6.939±0.023 | 6.778±0.278 |
> | Ours | 70.53±0.12 | 86.58±0.05| 79.35±0.06 | 5.862±0.030 | 6.224±0.051 | 6.947±0.118 |
>
> **Table 9: Performance comparison on MSD with SwinUNETR backbone.**
> | Method | Task02 Heart Average Dice | Task06 Lung Average Dice | Task09 Spleen Average Dice
> |:------|:-------:|:-------:|:-------:|
> | Scratch | 91.95±1.64 | 65.82±4.16 | 95.76±0.52
> | Full | 93.73±1.53 | 67.69±4.22 | 96.52±0.33
> | Ours | 95.84±1.26 | 78.09±4.04 | 97.06±0.40
>
> > [**Q4**]. The authors have clearly put a lot of effort in preparing the experiments and designing their solution but the numerous language issues throughout the papers make it very difficult to follow. There are many nominal statements without verbs (e.g : Secondly, improve the flexibility and lightness of the framework on p4) Some of the sentences are very convoluted and missing some words References to future results (section 4.2 referred to in section 3.2) does not help with clarity.
>
> [**A4**]. Thanks for your valuable advice, we have further improved certain inappropriate expressions in our manuscript, enhancing the writing quality of our paper.
> For example:
> | Before | After |
> |:------|:-------|
> |Secondly, improve the flexibility ... | Secondly, sole insertion in encoder part improves the flexibility of the whole framework. Our inserting strategy broadens the adaptability of Med-Tuning on visual foundation models while reducing tuned parameters.|
> |... results of Head in Sec. 4.2 show.  | Inadequate feature extraction will hinder performance even with a same robust decoder, as evidenced by the decline in results for Head tuning compared to Full fine-tuning, detailed in Tables 1, 2, and 3.|

---

> ### Author Response · Authors · 2024-03-16
>
> > [**Q5**]. The idea is quite interesting with comprehensive experiments but the overall clarity is poor and the conclusions may overstate the results. Statistical analysis is fully lacking and there is no clear link between the different pieces of the framework in terms of justification and/or explanation.
>
> [**A5**]. Here, we provide a summary of our detailed responses above:
> 1. We enhanced the overall clarity of the paper and improved certain inappropriate expressions.
> 2. We supplement some representative raw results accompanied by their standard deviations for statistical analysis, which demonstrate that there is no overstatement of our results.
>
> We hope the responses provided above address any concerns or questions you may have regarding our paper.

---

### Official Review · Reviewer_ufgr · 2024-03-08

**Confidence:** 4
**Preliminary Rating:** 4
**Recommendation:** Poster
**Final Rating:** 5

**Summary:**

This paper describes a parameter-efficient tuning (PET) method to adapt foundation models to the task of 3D medical image segmentation. They present three different contributions: 1) The Med-Tuning Framework, 2) The Med-Adapter Module, and 3) Extensive experiments on different modalities of medical imaging (MRI, CT) and ablation studies. In this framework, authors show the potential of harnessing pre-training done on foundation models to enhance segmentation performances and reduce training costs by only training parts of the model.

**Strengths:**

The paper has been well written with an extensive literature review and multiple experiments comparing their approach to other methods available and already published. Authors have shown improvements in previous results on all 3 datasets both in terms of Dice coefficient and Hausdorff distance.

**Weaknesses:**

Although the performances shown are superior to previous methods, I would have liked to see confidence intervals/standard errors of the performances as the results between the methods are very close. If there is an overlap of the results, one could not confidently conclude that such method performs best.
Another point that was a bit surprising to me was the lack of difference in performance shown in the ablation study. If I understood correctly the first line of Table 4 represents no presence of Med-Adapter, and the last line is your proposed method. Although the average values of your performances are superior, I wish the authors presented standard errors to ensure no to little overlap of the confidence intervals to effectively conclude on an improvement.
One last comment I would have is on the concept itself of PET. I would be curious to see how your method fares compared to training a nnUnet as an even simpler baseline. To my knowledge, nnUnet has always been shown to outperform other methods while being much lighter. Fine-tuning is an interesting process when very little data is present for training and testing but in the datasets you have here, such verification would be possible and empower your claim to use PET methods over retraining a model (albeit much lighter) from the ground up.

**Detailed Comments:**

Add Standard errors when presenting the results.
Add a comparison to a state-of-the-art nnUnet

**Justification Of Final Rating:**

The authors have addressed all my questions and interrogations. My final rating is therefore increased.
Indeed, some improvements seem to not be significant but interesting discussions can rise from this work.

**Justification Of The Preliminary Rating:**

The paper describes a method that is not innovative as shown by the extensive literature that exists already. It is however an interesting point of discussion for the community and extensive work has been done to compare their approach to existing methods and show improvements.

**Questions To Address In The Rebuttal:**

Cf weaknesses and detailed comments

**Special Issue:**

No

---

> ### Author Response · Authors · 2024-03-16
> **Sincerely appreciation for your recognition of our extensive literature review, experiments, and good writing**
>
> We sincerely appreciate your recognition of our extensive literature review, experiments, and good writing. We have addressed all of your questions in detail in the following Response section and will incorporate all feedback in the revision. We hope that our detailed explanations will give us the precious opportunity to raise the evaluation score of our work in your perspective.
>
> **Our Responses to Paper Weaknesses:**
>
> > [**Q1**]. Although the performances shown are superior to previous methods, I would have liked to see confidence intervals/standard errors of the performances as the results between the methods are very close. If there is an overlap of the results, one could not confidently conclude that such method performs best.
>
> [**A1**].
> Thanks for this helpful suggestion. Per your advice, we supplement some representative raw results accompanied by their standard deviations, which are listed below in Table 1-4 for your convenience to check. The results indicate that, despite **minor overlaps in certain cases**, our method is **generally comparable to or surpasses** the results of full fine-tuning and most of the previous PET methods.
>
> **Table 1: Performance comparison on KiTS 2019 with SwinUnet backbone.**
> | Method | Kidney Dice | Tumor Dice |
> |:------|:-------:|:-------:|
> | Scratch | 94.33±0.06 | 61.10±0.40 |
> | Full | 94.68±0.07 | 62.13±0.97 |
> | Head | 91.95±0.03 | 53.93±0.35 |
> | VPT-Shallow | 91.72±0.09 | 54.86±0.71 |
> | VPT-Deep | 91.53±0.12 | 53.41±0.76 |
> | Adapter | 93.02±0.04 | 57.15±0.62 |
> | AdaptFormer | 93.74±0.03 | 59.79±0.98 |
> | Pro-tuning | 90.34±0.08 | 51.19±0.46 |
> | ST-Adapter | 92.97±0.07 | 57.33±0.53 |
> | Ours | 95.69±0.05 | 70.14±0.54 |
>
> **Table 2: Performance comparison on BraTS2019 with SwinUnet backbone.**
> | Method | Dice(ET) | Dice(WT) | Dice(TC) | Hausdorff(ET)| Hausdorff(WT)| Hausdorff(TC)|
> |:------|:-------:|:-------:|:-------:|:-------:|:-------:|:-------:|
> Scratch|78.38±0.29|88.59±0.12|76.46±0.45|6.055±0.202|10.651±0.531|9.176±0.301|
> Full|78.26±0.25|89.56±0.07|79.16±0.63|4.327±0.598|6.149±0.179|6.704±0.120|
> Head|78.07±0.07|88.68±0.09|77.26±0.33|5.021±0.374|6.697±0.082|7.091±0.392|
> VPT-Shallow|77.16±0.18|88.30±0.15|76.77±0.27|5.421±0.108|6.154±0.044|7.345±0.053|
> VPT-Deep|77.02±0.22|88.65±0.07|76.91±0.32|5.297±0.220|7.088±0.157|7.940±0.132|
> Adapter|77.98±0.25|89.22±0.19|78.02±0.46|5.298±0.061|6.622±0.094|8.490±0.359|
> AdaptFormer|77.69±0.13|88.61±0.14|76.83±0.68|4.909±0.802|6.290±0.058|7.885±0.287|
> Pro-tuning|78.58±0.14|89.33±0.04|78.79±0.36|5.273±0.040|6.410±0.323|8.236±0.376|
> ST-Adapter|78.40±0.19|89.54±0.02|77.44±0.11|4.751±0.068|6.013±0.037|7.405±0.317|
> Ours|78.51±0.16|89.68±0.03|80.44±0.12|4.003±0.062|5.517±0.043|5.756±0.304|
>
> **Table 3: Performance comparison on BraTS2019 with ViT+UPerNet backbone.**
> | Method | Dice(ET) | Dice(WT) | Dice(TC) | Hausdorff(ET)| Hausdorff(WT)| Hausdorff(TC)|
> |:------|:-------:|:-------:|:-------:|:-------:|:-------:|:-------:|
> | Scratch | 64.96±0.30 | 83.03±0.17 | 71.34±0.39 | 7.635±0.162 | 10.602±0.504 | 10.942±0.186 |
> | Full | 68.49±0.20 | 85.56±0.05 | 75.12±0.56 | 6.672±1.492 | 7.878±0.155 | 10.525±1.091 |
> | Head | 65.71±0.04 | 84.19±0.08 | 74.77±0.29 | 6.128±0.298 | 7.505±0.036 | 7.864±0.324 |
> | VPT-Shallow | 66.02±0.11 | 84.72±0.14 | 75.84±0.28 | 6.114±0.087 | 7.506±0.020 | 8.471±0.014 |
> | VPT-Deep | 67.01±0.22 | 85.14±0.04 | 76.80±0.38 | 6.064±0.174 | 7.717±0.134 | 7.648±0.094 |
> | Adapter | 68.30±0.12 | 85.37±0.15 | 77.05±0.42 | 5.501±0.031 | 7.636±0.053 | 7.986±0.305 |
> | AdaptFormer | 65.88±0.16 | 84.34±0.14 | 74.77±0.62 | 6.652±0.568 | 8.204±0.030 | 8.430±0.232 |
> | Pro-tuning | 67.18±0.16 | 85.32±0.04 | 76.51±0.24 | 5.805±0.014 | 7.073±0.266 | 7.564±0.262 |
> | ST-Adapter | 69.18±0.19 | 86.27±0.04 | 79.18±0.07 | 6.077±0.043 | 6.939±0.023 | 6.778±0.278 |
> | Ours | 70.53±0.12 | 86.58±0.05| 79.35±0.06 | 5.862±0.030 | 6.224±0.051 | 6.947±0.118 |
>
> **Table 4: Performance comparison on MSD with SwinUNETR backbone.**
> | Method | Task02 Heart Average Dice | Task06 Lung Average Dice | Task09 Spleen Average Dice
> |:------|:-------:|:-------:|:-------:|
> | Scratch | 91.95±1.64 | 65.82±4.16 | 95.76±0.52
> | Full | 93.73±1.53 | 67.69±4.22 | 96.52±0.33
> | Ours | 95.84±1.26 | 78.09±4.04 | 97.06±0.40

---

> ### Author Response · Authors · 2024-03-16
>
> > [**Q2**]. Another point that was a bit surprising to me was the lack of difference in performance shown in the ablation study. If I understood correctly the first line of Table 4 represents no presence of Med-Adapter, and the last line is your proposed method. Although the average values of your performances are superior, I wish the authors presented standard errors to ensure no to little overlap of the confidence intervals to effectively conclude on an improvement.
>
> [**A2**]. Thanks for this helpful suggestion. Indeed, the first row of Table 4 (in our manuscript) represents the results without the Med-Adapter (i.e., full fine-tuning). Per your advice, we supplement results with standard deviations, which are listed below in Table 5 for your convenience to check. It can be observed that compared to full fine-tuning, Med-Tuning (default) has achieved improvements of **+0.44**, **+1.00**, **+3.18** in **Dice_ET**, **Dice_WT**, **Dice_TC**, and **-1.02**, **-1.18**, **-1.34** in **HF_ET**, **HF_WT**, **HF_TC**, respectively. Meanwhile, it has demonstrated smaller standard deviations in the results, indicating that our proposed Med-Tuning (default) exhibits lower randomness.
>
> **Table 5: Ablation study on the position of inserted Med-Adapter.**
> | n=0 | n=1 | n=2 | n=3 | Dice(ET) | Dice(WT) | Dice(TC) | Hausdorff(ET)| Hausdorff(WT)| Hausdorff(TC)| |
> |:------|:------|:------|:------|:-------:|:-------:|:-------:|:-------:|:-------:|:-------:|:-------:|
> -|-|-|-|78.07±0.28 |88.68±0.19 |77.26±0.40 |5.02±0.32 |6.70±0.51 |7.10±0.29
> √|√|-|-|74.83±0.30 |87.09±0.27 |72.94±0.44 |7.26±0.37 |13.12±0.60 |10.17±0.41
> √|√|√|-|75.60±0.31 |86.79±0.23 |73.41±0.39 |8.44±0.49 |12.32±0.53 |11.24±0.33
> √|√|√|√|78.51±0.27 |89.68±0.22 |80.44±0.34 |4.00±0.34 |5.52±0.54 |5.76±0.30 |(default)
>
>
> > [**Q3**]. One last comment I would have is on the concept itself of PET. I would be curious to see how your method fares compared to training a nnUnet as an even simpler baseline. To my knowledge, nnUnet has always been shown to outperform other methods while being much lighter. Fine-tuning is an interesting process when very little data is present for training and testing but in the datasets you have here, such verification would be possible and empower your claim to use PET methods over retraining a model (albeit much lighter) from the ground up.
>
> [**A3**]. Many thanks for this kind comment. It's worth noting that, on one hand, some compelling experiments in this paper have already been conducted using the nnU-Net training framework. As mentioned in the appendix, for the experiments on KiTS 2019 dataset, the data augmentations employed are in line with the prior work nnU-Net. In other words, the experiments carried out on the KiTS 2019 within this paper are based on the nnU-Net training framework. The results from experiments on KiTS 2019 clearly show that our method over various baselines (i.e., SwinUnet and ViT) under the nnU-Net framework have achieved significant improvements compared to other tuning methods. This represents a model performance enhancement that is independent of the training framework, sufficiently demonstrating the advantages of our method as well as the superiority of nnU-Net's data preprocessing and training framework.
>
> On the other hand, following your valuable suggestions, we have also endeavored, during the short time period of the rebuttal, to test the effectiveness of our Med-Tuning using nnU-Net as the baseline model. Based on the results obtained during this short period, the performance after combining nnU-Net with our Med-Tuning framework is very close to the reproduced performance of nnU-Net as the baseline model on KiTS 2019 dataset.
>
> Due to the constraints of time, we were unable to further enhance performance using our Med-Tuning. However, the comparative outcomes have sufficiently demonstrated the universality of our proposed method, without leading to any deterioration in performance. Moreover, there exists potential for achieving higher performance gains. We are optimistic about the future possibilities of our methodology and are committed to exploring these opportunities.

---

### Decision · Program_Chairs · 2024-04-06

Accept (Poster)